# New Data on the Distribution of Southern Forests for the West Siberian Plain during the Late Pleistocene: A Paleoentomological Approach

Anna A. Gurina [1,*], Roman Y. Dudko [1], Alexander V. Ivanov [2,†], Alexey A. Kotov [3], Yuri E. Mikhailov [4,5], Alexander A. Prokin [6], Alexander S. Prosvirov [7], Alexey Y. Solodovnikov [8,9], Evgenii V. Zinovyev [2] and Andrei A. Legalov [1,10,11]

1. Institute of Systematics and Ecology of Animals, Siberian Branch of Russian Academy of Sciences (RAS), Novosibirsk 630091, Russia
2. Institute of Plant and Animal Ecology, Ural Branch, Russian Academy of Sciences, Ekaterinburg 620144, Russia
3. A.N. Severtsov Institute of Ecology and Evolution, Moscow 119071, Russia
4. Department of Ecology & Nature Management, Ural State Forest Engineering University, Ekaterinburg 620100, Russia
5. Ecology Department, Ural Federal University, Ekaterinburg 620002, Russia
6. Papanin Institute for Biology of Inland Waters, Russian Academy of Sciences, Borok 152742, Russia
7. Faculty of Biology, Moscow State University, Moscow 119991, Russia
8. Zoological Museum (at the Natural History Museum of Denmark), 2100 Copenhagen, Denmark
9. Zoological Institute, Russian Academy of Sciences, St. Petersburg 199034, Russia
10. Department of Ecology, Biochemistry and Biotechnology, Altai State University, Barnaul 656049, Russia
11. Department of Forestry and Landscape Construction, Tomsk State University, Tomsk 634050, Russia
* Correspondence: auri.na@mail.ru
† This author has passed away.

**Abstract:** Subfossil remains of insects and branchiopod crustaceans (Cladocera and Notostraca) found in three late Pleistocene deposits in the Novosibirsk region in the vicinity of the village of Suzun have been described. The calibrated radiocarbon dates for these deposits were 24,893–25,966 cal BP (Suzun-1), 20,379–20,699 cal BP (Suzun-2), and 27,693–28,126 cal BP (Nizhny Suzun), which correspond to the onset of marine isotope stage 2 (MIS 2). The insect assemblages of these deposits are mainly represented by Coleoptera, which are noteworthy for high taxonomic and ecological diversity. At least 194 beetle species from 21 families have been found altogether. Of them, 74 species were found in the Pleistocene deposits of Western Siberia for the first time. All deposits were similar in species composition of beetles; Carabidae and Curculionidae prevailed everywhere. The ecological composition was dominated by steppe and tundra-steppe species; aquatic and riparian groups were also well represented. The Cladoceran and notostracan taxa revealed in Suzun-1 and Suzun-2 are characteristic of recent steppes rather than the forest zone of Western Siberia. The studied entomocomplexes are congruent with the periglacial "*Otiorhynchus*-type" fauna that inhabited the southern part of the West Siberian Plain at the end of the Pleistocene and had no close contemporary analogues. Cold and dry conditions, as well as the prevailing open landscapes of the tundra-steppe type, were the reconstructed conditions for this fauna. At the same time, the Suzun-1 and Suzun-2 entomocomplexes had a distinctive feature, namely a high proportion of forest species associated with both coniferous and deciduous trees. According to these data, at the beginning of MIS 2 in the Upper Ob region, spruce forests with the participation of small-leaved species (birch) were present. They were probably confined to river valleys and were not widely distributed.

**Keywords:** insecta; Coleoptera; Cladocera; Quaternary; Karginian interstadial; MIS 3; Sartanian stadial; MIS 2; forests

## 1. Introduction

Shifts of the natural biotic zones at the end of the Pleistocene ultimately led to the formation of the modern landscapes of the globe. Although tree pollen was recorded in the West Siberian Plain, even at the time of the last glaciation maximum, north temperate and boreal forests, which currently occupy most of this area, were absent or were at least much less developed here during marine isotope stage 3 (MIS 3) and MIS 2 [1–3]. A question about the degree of development of forests in northern Eurasia and about their flora and fauna is open to discussion.

Insects are very sensitive to changes in environmental conditions, and any changes in the diversity of this largest group of living multicellular organisms make it possible to detect even very slight changes in ecosystems [4–6]. Until recently, entomological studies were mainly carried out in northeastern Siberia [7–14]. Quaternary insects of the West Siberian Plain were also the objects of intensive studies [15–24].

The entomocomplexes of the end of MIS 3 and MIS 2, previously identified in the south of the West Siberian Plain, indicate the predominance of open landscapes of the tundra-steppe type with rare inclusions of the species confined to the forests [25–28]. This paper presents new data on insects from three deposits of the Upper Ob region located in the lower reaches of the Suzun River. According to these new data, the entomocomplexes found in these deposits include both the species characteristic of the late Pleistocene fauna associated with the open landscapes as well as a significant number of forest elements. The analysis of these entomocomplexes in this paper pays particular attention to the forest-associated species. In addition to insects, we analyzed some remains of the branchiopod crustaceans (Cladocera: Daphniidae and Notostraca: Triopsidae), which are also known as a valuable indicator group for the reconstruction of past ecological conditions [29–32].

## 2. Material and Methods

### 2.1. Study Area

The Suzun-1 (53.73169° N, 82.18172° E) and Suzun-2 (53.733336° N, 82.18352° E) deposits are located on the right bank of the Suzun River (right tributary of the Ob River), 2.5 km upstream from the mouth (Figure 1). The Nizhny Suzun locality (53.71668° N, 82.12691° E) is located on the right bank of the Ob River, 1.7 km downstream of the mouth of the Suzun river (about 4 km from Suzun-1 and Suzun-2). Our study region, which is not disturbed by agricultural activities, is presently covered with pine tree grassy-shrub forests. In the floodplains of the area, the vegetation is represented by shrubs and meadows [33]. The forest-steppe of Western Siberia is located within the continental sector of the temperate latitudinal climatic zone [34,35]. The average temperatures in January are −19 to −20 °C, and the average temperatures in July are +18 to +19 °C [35].

### 2.2. Methods and Geological Settings

Sampling was carried out according to the method of Coope [36] with subsequent modifications [27,37,38]. Only relatively well-preserved "potentially identifiable" fragments of insects (such as whole or halves of head capsules, elytra, pronotum, terminalia, etc.) were used for the analysis. The identification was carried out by comparing fragments with modern material. For comparison, the collections of the following institutions were used: Institute of Systematics and Ecology of Animals, Siberian Branch of RAS (Novosibirsk); Institute of Plant and Animal Ecology, Ural Branch of RAS (Ekaterinburg); Zoological Institute of RAS (St. Petersburg); Papanin Institute for Biology of Inland Waters, RAS (Borok, Yaroslavl Oblast); Paleontological Institute of RAS (Moscow); Moscow M.V. Lomonosov State University; Moscow Pedagogical State University; and the Zoological Museum at the Natural History Museum of Denmark (Copenhagen). The age was determined by the radiocarbon method performed at the A. Herzen Russian State Pedagogical University in St. Petersburg using the method of M.A. Kulkova. Radiocarbon age calibration was carried out using the Calib Rev 8.1.0 program (UK) with the IntCal20 curve, and a precision of 1δ.

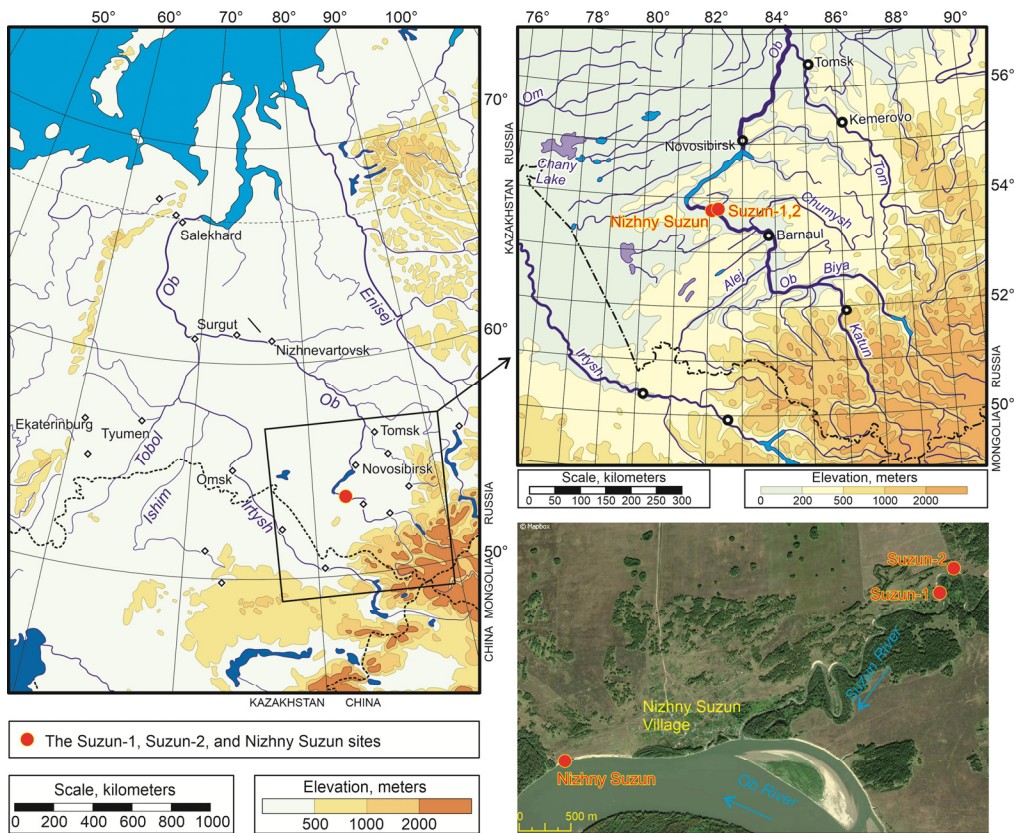

**Figure 1.** The study area within the maps of the West Siberian Plain, the southeast of Western Siberia, and the image of the Mapbox.

To estimate the number of individuals of each species in the samples, the Nmin indicator (minimum number of individuals) was used. Nmin was calculated based on the assumption that those parts of the exoskeleton that could belong to one insect actually belonged to one individual. Numerically, for each beetle species, it is equal to the maximum number of fragments of one type (head, pronotum, or left or right elytron).

When comparing the species composition, the Szymkiewicz–Simpson pairwise similarity coefficient was used: Ks = c/min (a,b) × 100%, where a and b are the numbers of species in the first and second samples, respectively, and c is the number of common species in these samples [39,40].

Descriptions of each deposit were compiled and are given in Tables 1–3.

**Table 1.** Description of the Suzun-1 section.

| Layer No. | Depth of Bedding, m | Thickness, m | Description |
|---|---|---|---|
| 1 | 0.0–0.3 | 0.3 | Modern soil. |
| 2 | 0.3–2.3 | 2.0 | Light brown sandy loam. Dense, dry, porous, with spots of ferrugination. |
| 3 | 2.3–2.4 | 0.1 | Gray loam. Dense. |
| 4 | 2.4–3.2 | 0.8 | Brown sandy loam. Dense, with ferrugination along the roots of trees. |
| 5 | 3.2–4.5 | 1.3 | Gray-brown loam with spots of ferrugination. |
| 6 | 4.5–6.1 | 1.6 | Horizontal interbedding of medium-grained light brown sands and loams with ferruginous inclusions. |
| 7 | 6.1–6.9 | 0.8 | Alternation of dark brown sandy loams and loams with inclusions of plant detritus. |
| 8 | 6.9–7.85 | 0.95 | Blue-gray clays with lenses and layers of alluvial plant detritus. Five samples were taken for entomological analysis (Table 4). |
| 9 | 7.85–8.25 | 0.4 | Gray sand. Medium-grained, moisture-saturated, goes under the water level. |

**Table 2.** Description of the Suzun-2 section.

| Layer No. | Depth of Bedding, m | Thickness, m | Description |
|---|---|---|---|
| 1 | 0.0–0.5 | 0.5 | Modern soil. |
| 2 | 0.5–5.5 | 5.0 | Light brown sandy loam. Dense, dry, porous, with spots of ferrugination. |
| 3 | 5.5–12.0 | 6.5 | Gray loam. Dense, interbedded with brownish-gray sandy loam. |
| 4 | 12.0–13.3 | 1.3 | Brown sandy loam. Dense, with sparse lenses and sublayers of alluvial detritus. |
| 5 | 13.3–16.0 | 2.7 | Bluish-gray sandy loam. Dense, with lenses and sublayers of alluvial detritus. Four samples were taken for entomological analysis (Table 4). |
| 6 | 16.0–16.5 | 0.5 | Gray sand. Medium-grained, moisture-saturated, goes under the water level. |

**Table 3.** Description of the Nizhny Suzun section.

| Layer No. | Depth of Bedding, m | Thickness, m | Description |
|---|---|---|---|
| 1 | 0.0–0.6 | 0.6 | Modern soil. |
| 2 | 0.6–1.6 | 1.0 | Sandy loam. Dense, dry, porous, with spots of ferrugination. |
| 3 | 1.6–3.1 | 1.5 | Light gray sands. Cross-bedded, medium- and coarse-grained. |
| 4 | 3.1–4.5 | 1.4 | Light gray sands. Horizontal-bedded, medium- and coarse-grained, with sublayers of cross-bedded sands. |
| 5 | 4.5–4.6 | 0.1 | Sublayer of buried soil. |
| 6 | 4.6–5.1 | 0.6 | Brownish sands. Horizontal-bedded, medium- and coarse-grained, with spots and sublayers of ferrugination. |
| 7 | 5.1–10.0 | 4.9 | Light-gray sands. Horizontal-bedded, medium- and coarse-grained, with spots and sublayers of ferrugination. |
| 8 | 10.0–11.5 | 1.5 | Dark brown sandy loam. Wet. |
| 9 | 11.5–12.0 | 0.5 | Bluish-gray clay with sublayers of alluvial plant detritus. Two samples were taken for entomological analysis (Table 4). |

**Table 4.** Coordinates of the sections, and the depths and radiocarbon dates of the samples.

| Section | Coordinates | Samples: Depth, m | Radiocarbon Date, BP | Laboratory Code | Calibrated Age, cal BP |
|---|---|---|---|---|---|
| Suzun-1 | N53.73169°; E82.18172° | S1: 6.9–7.1 S2: 7.1–7.25 S3: 7.25–7.45 S4: 7.45–7.65 S5: 7.65–7.85 | S1: 21,190 ± 500 | SPb_3011 | 24,893–25,966 |
| Suzun-2 | N53.73334°; E82.18352° | S1: 13.3–13.6 S2: 13.6–13.8 S3: 14.1–14.3 S4: 14.5–14.7 | S2: 16,984 ± 120 | SPb_3125 | 20,379–20,699 |
| Nizhny Suzun | N53.71668°; E82.12,691° | S1: 11.5 –11.7 S2: 11.7–11.95 | S2: 23,737 ± 200 | SPb_3126 | 27,693–28,126 |

*2.3. Material*

The insect subfossil remains were found in only one layer in each section, namely the bluish-gray clays at the base of the Suzun-1 and Nizhny Suzun sections and the bluish-gray sandy loams in the lower part of the Suzun-2 section. In total, five samples were collected from Suzun-1, four samples were collected from Suzun-2, and two samples were collected from the Nizhny Suzun sites by A.A. Legalov, E.V. Zinoviev, R.Yu. Dudko, E.R. Dudko, A.A. Gurina, and M.S. Kireev during 16–19 August of 2014. Radiocarbon dating was carried out based on plant detritus from the samples. The ages of all sections correspond to the first half of MIS 2 (Table 4).

## 3. Results

### 3.1. Taxonomic Composition

Five samples from the Suzun-1 deposit were processed. In total, 822 fragments were extracted and identified. These fragments belonged to at least 484 individual insects (Insecta) and two spiders (Aranei) (Appendix A Table A1). The Suzun-1 taphocenosis was mainly represented by beetles (Coleoptera), which formed 97% of individuals (Figures 2 and 3). Diptera, Hemiptera, and Hymenoptera were only represented here by singletons. The Coleoptera of Suzun-1 accounted for at least 145 species from 18 families. Of them, according to Nmin (the minimum number of individuals), the most numerous were weevils (Curculionidae), which accounted for 29% of individuals, ground beetles (Carabidae) with 24% of individuals, and rove beetles (Staphylinidae) with 12% of individuals. Other beetle families (Scarabaeidae, Tenebrionidae, Chrysomelidae, Silphidae, Scolytidae, Byrrhidae, and Dytiscidae) only accounted for 2% to 4% of individuals. The families Helophoridae, Brentidae, Leiodidae, Elateridae, Hydraenidae, Heteroceridae, Malachiidae, and Cerambycidae were only represented in this deposit by singletons. Few ephippia (modified moultingexivia with resting eggs) of *Daphnia* (*Ctenodaphnia*) *magna* Straus (Cladocera: Daphniidae) and few mandibles of a tadpole shrimp (Notostraca: Triopsidae) were found in this deposit.

From the Suzun-2 deposit, four samples were processed, from which 346 fragments belonging to 264 individuals of beetles (Coleoptera) and two individuals of true bugs (Hemiptera) were obtained. Here, Coleoptera were represented by 101 species from nineteen families, of which the most numerous (by Nmin) were ground beetles (Carabidae), which accounted for 34% of individuals, and weevils (Curculionidae), with 26% of individuals (Figure 3). Exemplars of rove beetles (Staphylinidae) and leaf beetles (Chrysomelidae) accounted for 8% and 7%, respectively. The remaining beetle families (Silphidae, Scarabaeidae, Tenebrionidae, Dytiscidae, Helophoridae, Elateridae, Brentidae, Scolytidae, Hydrophilidae, Hydraenidae, Leiodidae, Byrrhidae, Meloidae, Heteroceridae, Malachiidae, and Cerambycidae) were only represented by singletons. Relatively numerous ephippia of the *Daphnia* (*C.*) *magna* and *D.* (*Daphnia*) *pulex* group were also found in this deposit (Figure 4).

Two samples were processed from the Nizhny Suzun deposit. In total, only 12 (sample S1) and 26 (sample S2) insect fragments, all belonging to the order Coleoptera, were extracted (Figure 5). The families Curculionidae (eight species) and Carabidae (nine species) were predominant here. The families Scarabaeidae, Byrrhidae, Silphidae, Tenebrionidae, and Chrysomelidae were represented by one or two species. Most of the species in the Nizhny Suzun taphocenosis were represented by 1–2 fragments. Thus, we assume that the sample that we were able to obtain represents only a small fraction of the real local fauna of that time (27.6–28.0 ka BP). No remains of branchiopod crustaceans were found in this locality.

The entire list of the species from all three deposits includes at least 194 beetle species from 21 families. Of them, 137 taxa were identified to the level of species or a species group. In this list, 74 species of beetles (marked with an asterisk in Appendix A Table A1) and at least three taxa of the branchiopods are reported for the first time from late Pleistocene deposits of southern Western Siberia.

### 3.2. Comparison of the Species Composition

To compare the species composition of the entomocomplexes and individual samples, the Szymkiewicz–Simpson pairwise similarity coefficient was used. When comparing the species compositions of the samples from the Suzun-1 deposit, their high level of similarity (41–58%) with each other was revealed. Samples from the Suzun-2 deposit were also highly similar to each other (45–68% similarity) (Figure 6). The two samples from the Nizhny Suzun entomocomplex were 17% similar to each other due to their small sizes and one shared species, the *Otiorhynchus obscurus* weevil.

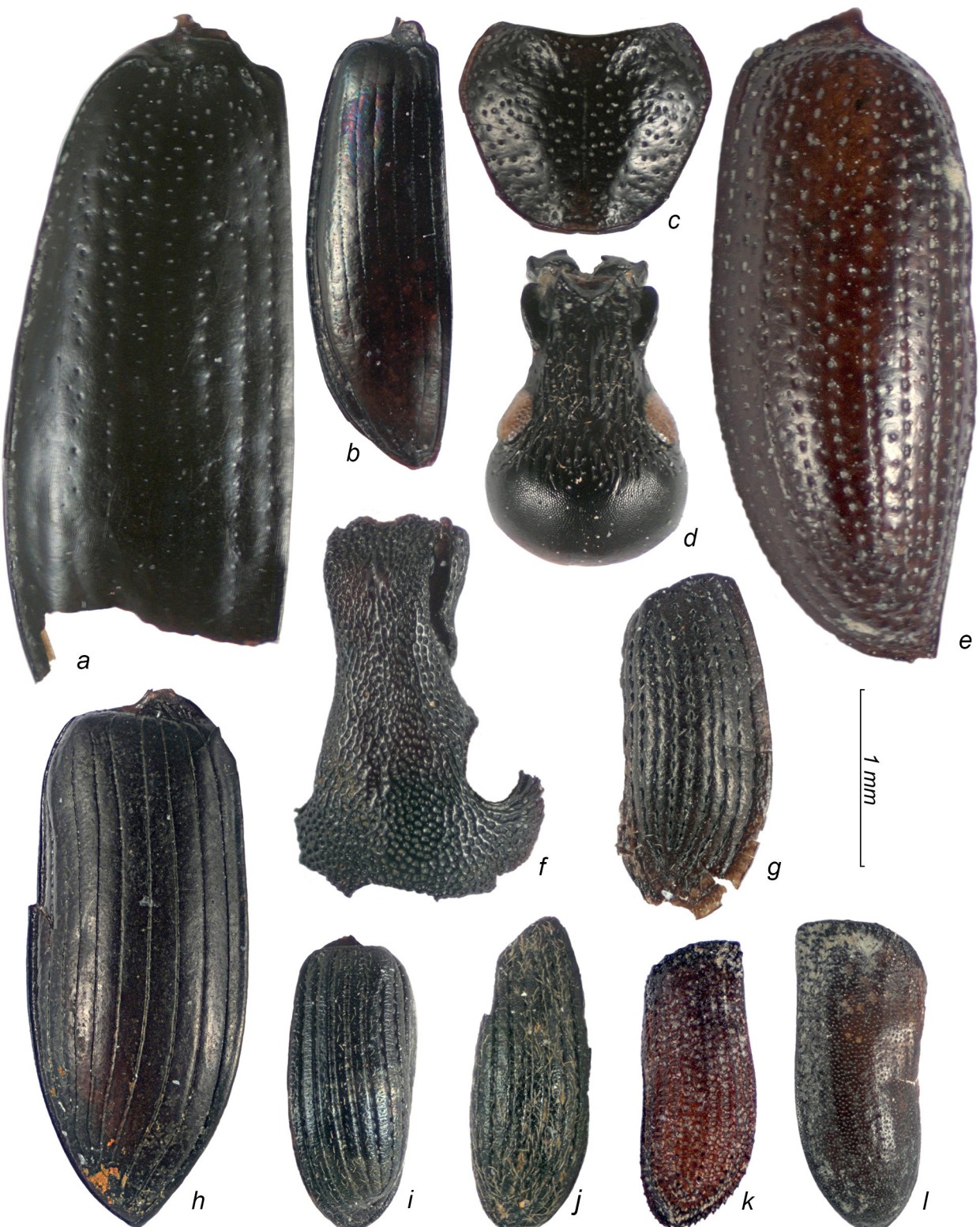

**Figure 2.** Carabidae (**a–c**), Curculionidae (**d–g**), Scarabaeidae (**h**), Brentidae (**i,j**), and Scolytidae (**k,l**) fragments from the Suzun-1 site. (**a**) *Diacheila polita* (S1), (**b**) *Bembidion* cf. *roborovskii* (S1), (**c**) *Cymindis* cf. *kasakh* (S5), (**d**) *Otiorhynchus subocularis* (S5), (**e**) *O.* af. *ursus* (S5), (**f**) *Trichalophus biguttatus* (S5), (**g**) *Anoplus plantaris* (S4), (**h**) *Aphodius plagiatus* (S5), (**i**) *Omphalapion hookerorum* (S5), (**j**) *Trichapion simile* (S4), (**k**) *Phloeotribus spinulosus* (S4), (**l**) *Polygraphus subopacus* (S4). (**a–b**), (**e**), (**g–l**) elytra; (**c**) pronotum; (**d,f**) head. Scale bar: 1 mm.

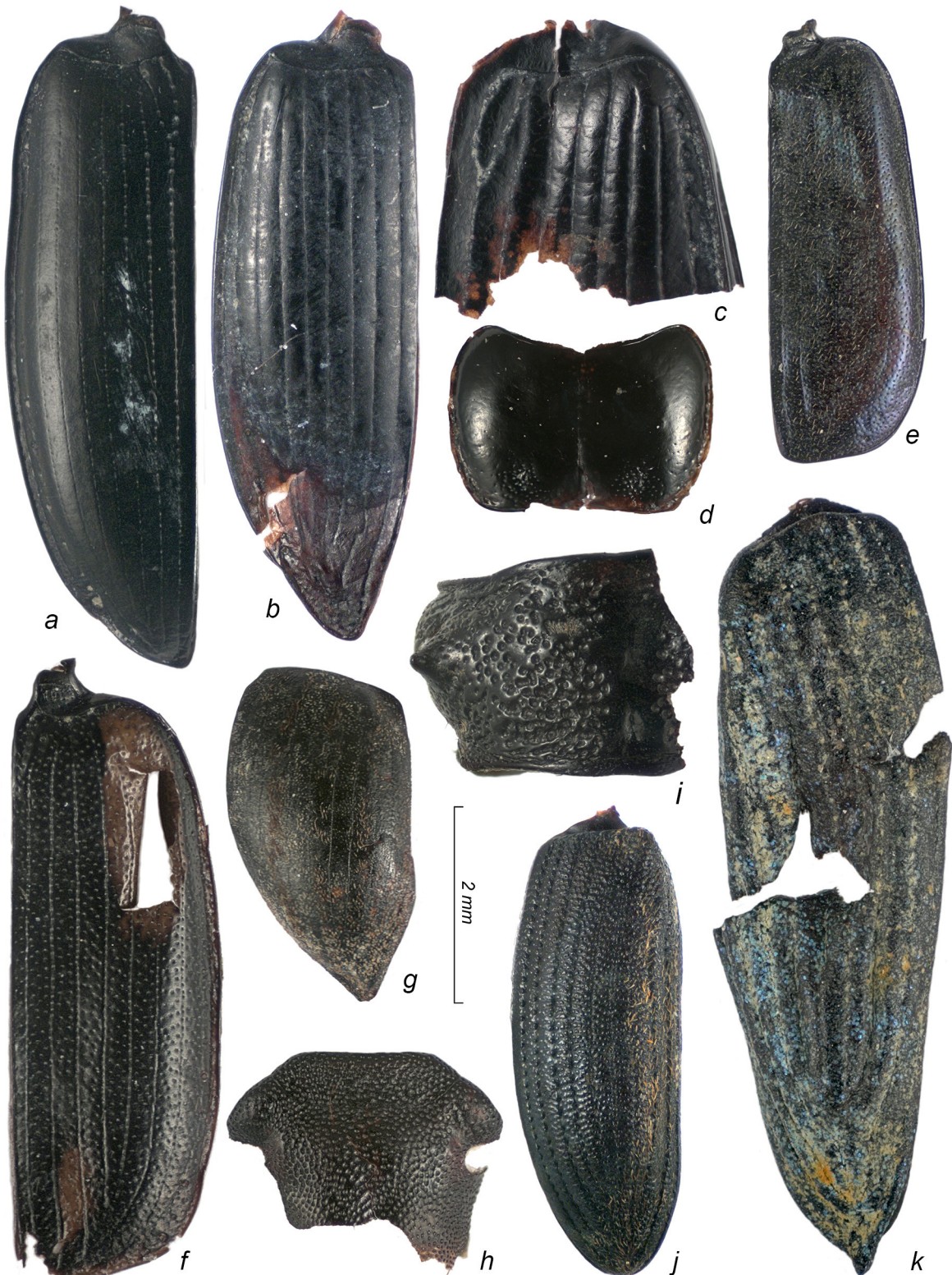

**Figure 3.** Carabidae (**a**–**f**), Byrrhidae (**g**), Tenebrionidae (**h**), Cerambycidae (**i**), fragments from the Suzun-1 site and Curculionidae (**j**–**k**) fragments from the Suzun-2 site. (**a**) *Poecilus* cf. *ravus* (S2), (**b**) *Pterostichus altainus* (S3), (**c**) *P. maurusiacus* (S3), (**d**) *Harpalus amputatus* (S5), (**e**) *Lebia punctata* (S2), (**f**) *Cymindis binotata* (S3), (**g**) *Porcinolus murinus* (S5), (**h**) *Platyscelis* sp. (S3), (**i**) *Eodorcadion carinatum* (S2), (**j**) *Tournotaris bimaculata* (S1), (**k**) *Chlorophanus sibiricus* (S1). (**a**–**c**), (**e**–**g**), (**j**–**k**) elytra; (**d**,**i**) pronotum; (**h**) head. Scale bar: 2 mm.

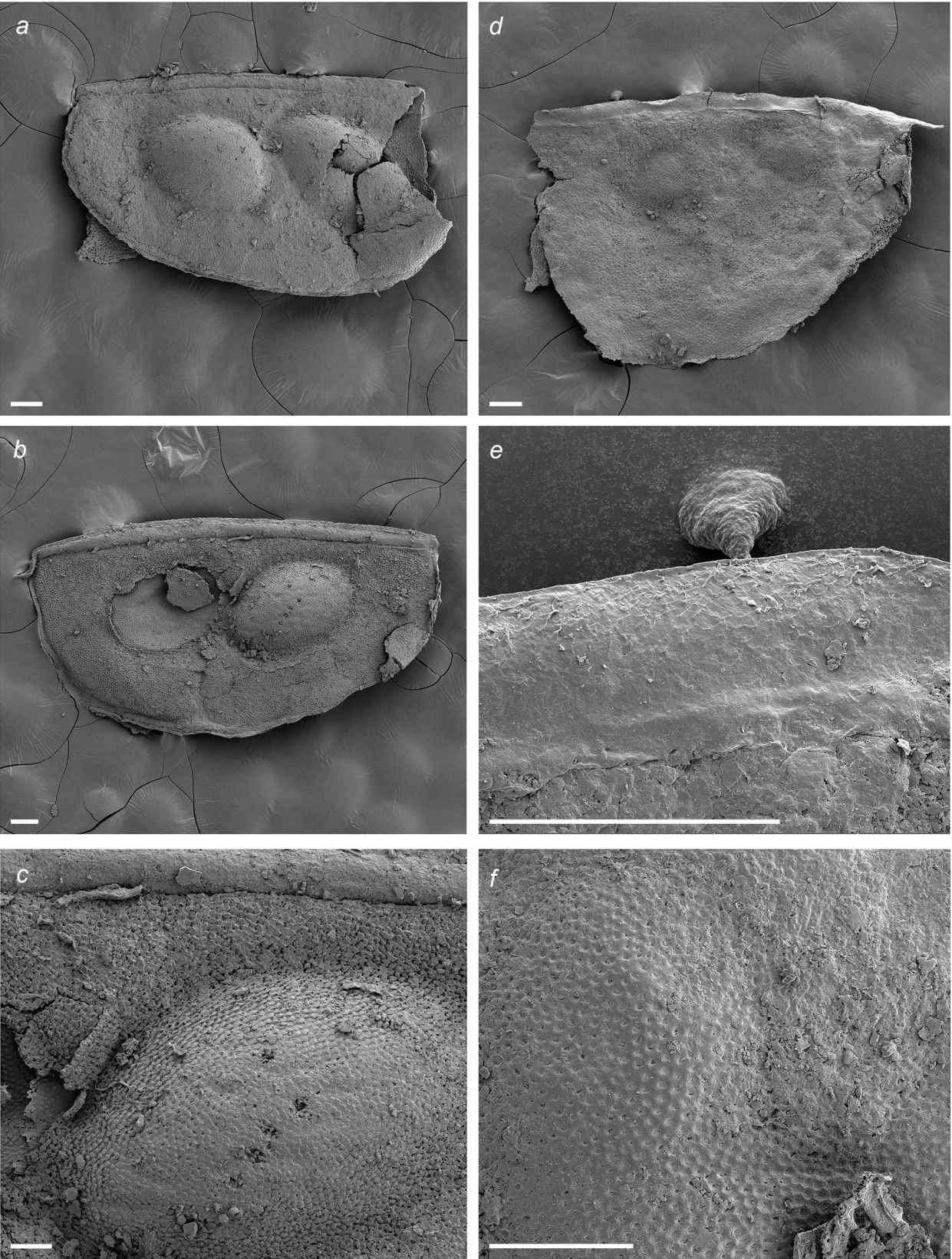

**Figure 4.** Ephippia of the *Daphnia* (*Ctenodaphnia*) *magna* (**a**–**c**) and *D.* (*Daphnia*) *pulex* group (**d**–**f**) from the locality Suzun-2. All scale bars are 0.1 mm.

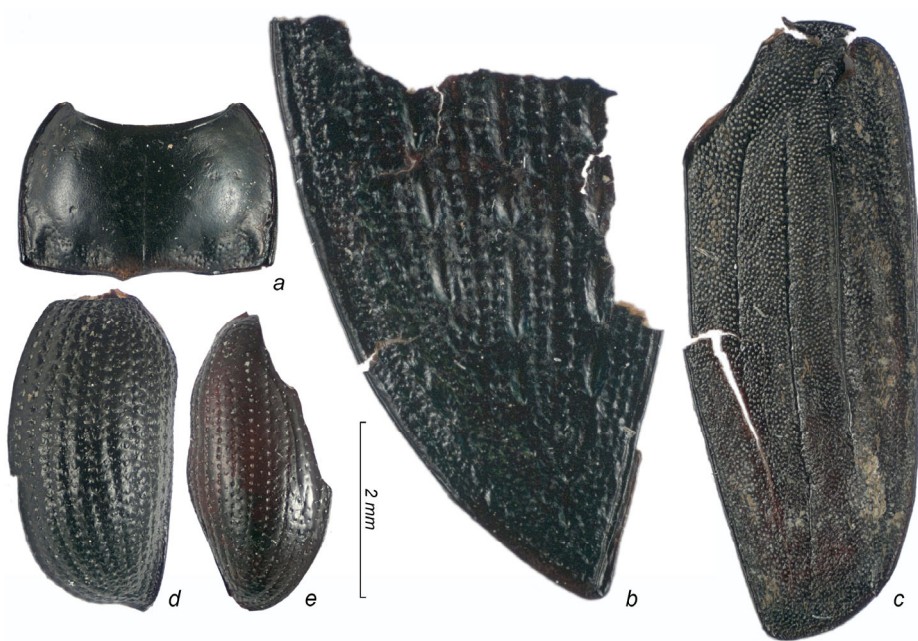

**Figure 5.** Carabidae (**a**,**b**), Silphidae (**c**), and Curculionidae (**d**,**e**) fragments from the Nizhny Suzun site. (**a**) *Amara quenseli* (S2), (**b**) *Carabus arvensis* (S1), (**c**) *Aclypea opaca* (S2), (**d**) *Otiorhynchus pullus* (S2), (**e**) *O. unctuosus* (S2). (**a**) pronotum, (**b**–**e**) elytra. Scale bars: 2 mm.

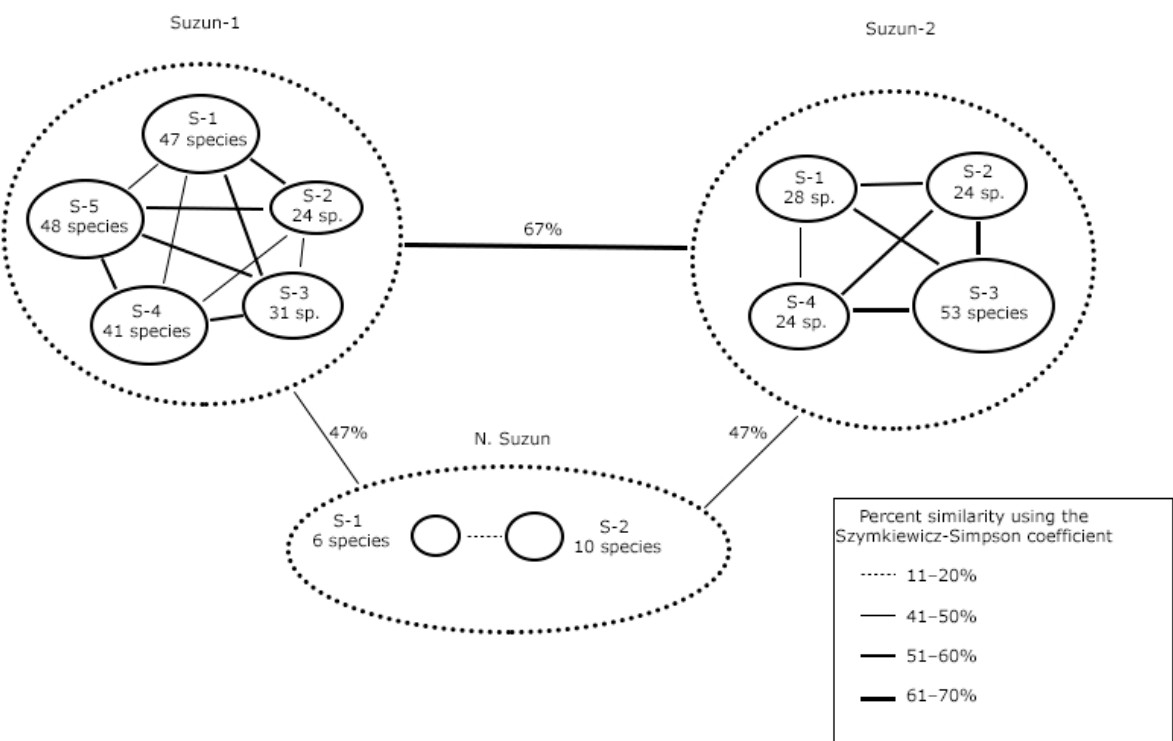

**Figure 6.** Similarity graph of insect assemblages between the samples and the sites of Suzun-1, Suzun-2, and Nizhny Suzun.

Despite the significant difference in age between the Suzun-1 and Suzun-2 taphocenoses, a comparison between them showed a 67% similarity of their entomocomplexes. A high level of similarity between them was also revealed in the composition of the branchiopod species complexes. Finally, there was a high level of similarity between the individual samples from each deposit (Suzun-2 S4 and Suzun-1 S1—62%; Suzun-2 S4 and Suzun-1 S5—66%; Suzun-2 S3 and Suzun-1 S3—66%; and Suzun-2 S2 and Suzun-1 S4—63%). The

entomocomplex of Nizhny Suzun was 47% similar with the entomocomplexes of Suzun-1 and Suzun-2 (Figure 6).

## 4. Discussion

### 4.1. Modern Distribution of Species

At present, about half of the Coleoptera species (Suzun-1—60%, Suzun-2—59%, and Nizhny Suzun—46%) that were found in the respective taphocenoses are present in the recent fauna of the study region. Other species also occur in the recent fauna, but they have shifted their distribution ranges to the north, south, east, or even west.

In the Suzun-1 and Suzun-2 beetle assemblages, 12–13% of the species (*Pelophila borealis*, *Diacheila polita*, *Otiorhynchus politus*, *Polydrusus amoenus*, *Sitona ovipennis*, *Agabus adpressus*, *A. coxalis*, and *Helophorus obscurellus*) currently live to the north of the studied region, in the taiga and tundra zones [41,42]. In total, 12% of the species (*Harpalus amputatus*, *Stephanocleonus suvorovi*, and *Helophorus parajacutus*) have shifted their ranges to the east and probably prefer a drier and sharply continental climate [42,43]. In total, 13–16% of the species (*Bembidion almum*, *Mylabris ledebouri*, and *Aclypea calva*) currently are distributed to the south of the study region [25,44]. In addition, the Suzun-2 taphocenosis contains species that today live to the west of the study area (*Cymindis* cf. *kasakh* and *Bembidion* cf. *aeneum*) [45]. In the Lower Suzun entomocomplex, in addition to the species currently living in this area, 46% (*Amara* cf. *saginata* and *Otiorhynchus obscurus*) are species now distributed to the south of the study area, and one species (*Amara quenseli*) is now shifted to the north [25,46].

The faunistic complex of branchiopods revealed for the examined deposit was one that formed in the water bodies of the arid belt millions of years ago (31). This complex was very typical for modern water bodies of the steppes of southern Western Siberia and Northern Kazakhstan [47].

### 4.2. Ecology

Based on the ecological preferences of the species found in the Suzun-1, Suzun-2, and Nizhniy Suzun deposits, we can draw a conclusion about the past environmental conditions in the area of the deposits.

Most of the Suzun-1 and Suzun-2 entomocomplexes were species characteristic of the open landscapes of the steppe (*Amara rupicola*, *Harpalus salinus*, *Baris lepidii*, *Eremochorus steppensis*, *Paophilus albilaterus*, and *Sitona obscuratus*) and tundra-steppe types (*Otiorhynchus* af. *ursus*, *Trichalophus biguttatus*, *Sitona ovipennis*, *Stephanocleonus foveifrons*, *S. suvorovi*, *Bembidion dauricum*, and *Diacheila polita*) (Figure 7). Among aquatic beetles, there were also representatives of the steppe (*Helophorus parajacutus*) and tundra-steppe (*Agabus adpressus*, *A. coxalis*, and *Helophorus obscurellus*) complexes. Species biologically associated with the plant families Chenopodiaceae (*Eremochorus steppensis*, *Baris artemisiae*, *Otiorhynchus obscurus*, and *Stephanocleonus foveifrons*) and Brassicaceae (*Phyllotreta nemorum*, *Colaphellus alpicola*, and *Aulacobaris lepidii*) was widely represented in the taphocenosis.

The intrazonal complex of insects associated with the banks of water bodies is quite diverse. It includes species that live on the banks of rivers (*Nebria gyllenhali*, *Dyschiriodes tristis*, and *Bembidion obliquum*) and species developing on aquatic and semiaquatic vegetation (*Bagous* spp., *Notaris scirpi*, *Phytobius leucogaster*, *Chlorophanus sibiricus*, and *Tournotaris bimaculata*). Species confined to meadow habitats (*Polydrusus amoenus* and *Phyllobius pomaceus*), including those that develop on legumes (*Tychius quinquepunctatus*, *Sitona obscuratus*, and *S. ovipennis*), are also widely represented in this complex.

Moreover, truly aquatic animals are also well represented in the complex. The Suzun-2 deposit was especially rich in such species, which undoubtedly suggests the existence of "steppe-type" water bodies in this area back then [32,48]. However, the "maximum watering" of this deposit, which can be assumed from the presence of the maximum number of crustacean remains, may also reflect the lower position of this paleohabitat in the relief, its proximity to a large river, or local microclimatic conditions.

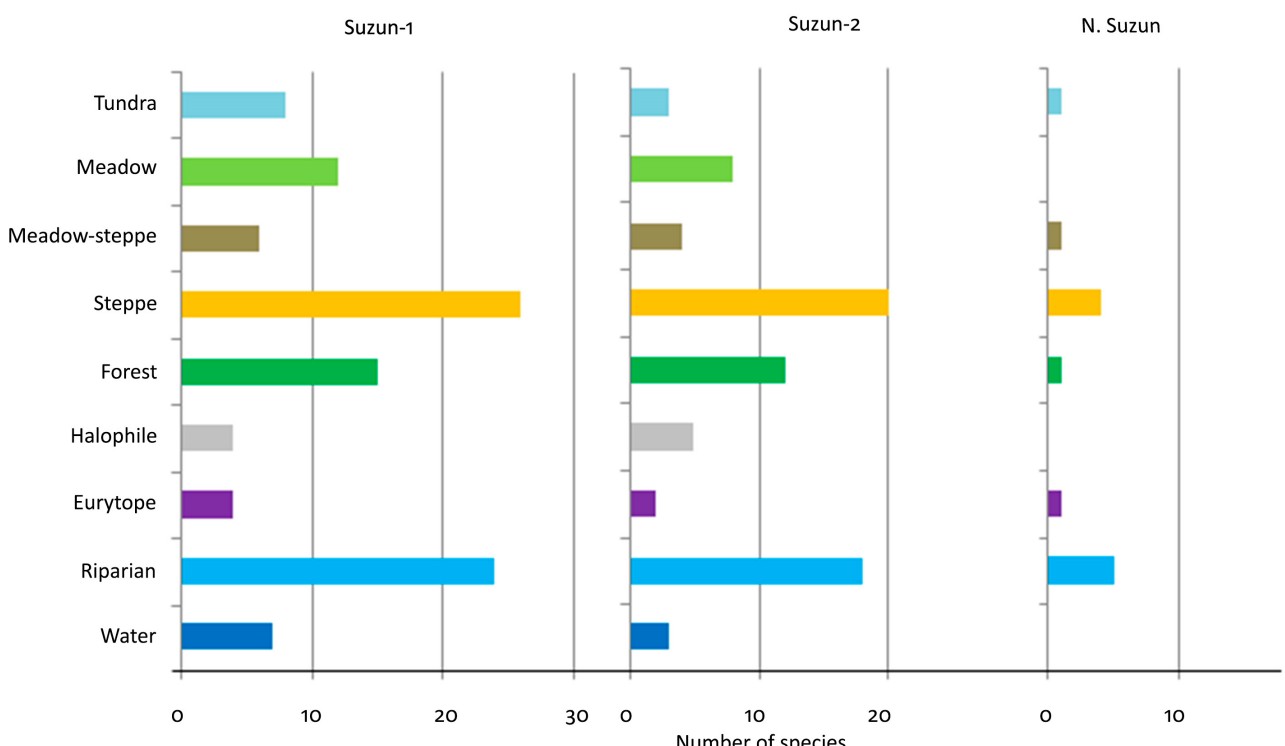

**Figure 7.** Ecological compositions of Coleoptera from the Suzun-1, Suzun-2, and Nizhny Suzun sites.

The presence in the Suzun-1 and Suzun-2 taphocenoses of species associated with woody vegetation is worth special consideration. In the Suzun-1 taphocenosis, 14 species now associated with forest biotopes were found. Among them there were two species of bark beetles, *Phloeotribus spinulosus* and *Polygraphus subopacus* (Figure 2k,l), associated with coniferous trees, mainly spruce; the weevils *Trichapion simile* and *Anoplus plantaris* (Figure 2g,j), which develop on birches and other small-leaved trees (*Eudipnus mollis*, *Phyllobius crassipes*, and *Phyllobius virideaeris*); and subcortical (*Phloeostiba lapponica*) or forest floor (*Olophrum fuscum*) rove beetles.

In Suzun-2, nine species of beetles now common in forests were found. In addition to the bark beetle *Phloetribus spinulosus*, there was a weevil species that lives in small-leaved forests or grasslands (*Otiorhynchus pullus*) as well as the forest weevil *Phyllobius virideaeris* and the rove beetle *Tachinus rufipes*.

Only one species of ground beetle, *Carabus arvensis* (Figure 5b), characteristic of dry forests [49], was found in the entomocomplex of the Nizhny Suzun. The basis of this taphocenosis was formed by species characteristic of open landscapes of the steppe type. These were weevil species of the genus *Otiorhynchus* developing on Asteraceae as well as species (*Tournotaris bimaculata* and *Notaris scirpi*) living on various riparian plants: *Carex*, *Typha latifolia*, *Glyceria*, and *Phalaroides*. Of the cold-loving species, the presence of *Patrobus* cf. *septentrionis* and *Amara quenseli*, which are currently characteristic of the tundra zone but have wider ecological preferences [49,50], should be noted.

Based on these data, we can conclude that the assemblage of forest beetles in the Suzun-1 taphocenosis was more diverse than that in the Suzun-2 deposit, which could be explained by the older age of the former. According to the beetle assemblages of both localities, forests were present there, probably with a predominance of spruce. A noticeable presence of small-leaved tree species, at least birch, follows from the composition of the Suzun-1 entomocomplex.

The absence of insects obligately associated with small-leaved forests in the beetle assemblage of Suzun-2 may be due to incomplete data in the deposit or a consequence of the deterioration of conditions that led to the reduction in birch forests. This conclusion is

consistent with the larger number of subfossil remains of cladocerans, which now mainly live in the steppe zone and were found in abundance in the Pleistocene tundra-steppes [30].

An assessment of the trophic preferences of the species found in the entomocomplexes from all three localities showed that they contained carnivorous and herbivorous insects in equal proportions. On average, each of these groups made up more than 40% of the entomocomplexes. In addition, in each taphocenosis there were sapro- and detritophages, which made up 6–7% of the entomocomplexes, and mixophytophages, which made up no more than 1%.

### 4.3. Comparison with the Region Entomocomplexes of a Similar Age

In the south of the West Siberian Plain, the end of MIS 3 corresponded to two previously studied entomological assemblages, Kalistratikha (28.3–29.1 ka BP) [26] and Kizikha-2 (30.0–30.8 ka BP) [28]. The taphocenoses of Suzun-1 and Suzun-2 showed a fairly high similarity with the Kizikha-2 entomocomplex (50%), which was probably due to the small sample in Kizikha-2, represented mainly by meadow species. The similarities of the taphocenoses of Suzun-1 and Suzun-2 with the Kalistratikha entomocomplex were much lower and amounted to 34% and 24%, respectively. In this taphocenosis, high proportions of steppe and meadow species were revealed, indicating conditions that were drier and colder than the modern conditions but warmer and wetter than during the subsequent MIS 2. The landscapes were characterized as dry steppes with meadow and bush (willow) vegetation in the depressions of the relief (floodlands).

The taphocenoses at Suzun river showed moderate (20–30% with Kizikha-1 and Dubrovino) or even a significant (40–50% with Bunkovo) similarities with the MIS 2 entomocomplexes in the south of the West Siberian Plain. The similarity was mainly due to the high proportion of riparian and steppe species (*Patrobus* cf. *septentrionis*, *Tournotaris bimaculata*, *Otiorhynchus* spp., and *Stephanocleonus* spp.), which were characteristic of the open landscapes of the late Pleistocene.

Thus, the entomocomplexes from the Suzun river were quite close to all entomocomplexes of close ages studied in the region. The similarities were even more obvious when the ecological compositions and modern distributions of subfossil species were taken into account. Steppe species, including halobionts, were well represented and often dominated everywhere. Tundra species were also well represented. Most of the species from taphocenoses are absent in the modern regional fauna and are distributed in more northern, southern, or eastern regions. Usually, species of the genus *Otiorhynchus*, such as *O.* af. *ursus* or closely related *O. bardus,* are predominant. All these are characteristic features of the so-called "*Otiorhinchus*-type" fauna, which has no close modern analogues and was the common fauna in MIS 3 and MIS 2 in the southern part of the West Siberian Plain and Ural foreland [28,45].

At the same time, a high proportion of the species obligately associated with trees or otherwise living in the forest communities was a distinctive feature of the entomological complexes of the Suzun-2 and, especially, Suzun-1 deposits. In other taphocenoses of the south of the West Siberian Plain, forest insect species were either absent or singly recorded [25–28]. This indicates that the conditions in the Ob River basin at the mouth of Suzun river at the onset of MIS 2 were different from the rest of the south of the West Siberian Plain. The forest areas reconstructed there were probably isolated and not widely distributed.

### 4.4. Paleobotanical and Theriological Landscape Reconstructions for the Region in MIS 3 and MIS 2

The Ob River basin in the Novosibirsk region has repeatedly attracted the attention of researchers of the Quaternary period. The theriological data for the Novosibirsk Ob region at the end of the Karginsky interstadial suggest steppe landscapes with the presence of subarctic species (reindeer and musk ox) and the complete absence of forest species (elk and red and giant deer) [51,52]. According to the data on small mammals, for this time, in

deposits south of 56° N, forest-steppe and steppe faunas were recorded, in which there were no lemmings. These faunas also reflect the spread of open treeless landscapes [53].

In the vicinity of the Nizhny Suzun, a palynological analysis of the late Pleistocene layers was carried out in the 1970s. M.R. Votakh showed a change in the composition of vegetation from MIS 3 to the end of MIS 2 [54]. In the composition of vegetation in the deposits dated 33,600 ± 2400 BP, a 67–91% predominance of tree species, represented by birch, pine, cedar, and spruce, was established. Among the herbaceous plants, there were forbs, grasses, quinoa, wormwood, and Compositae. Above these layers, in deposits dated 28,000 ± 6200 BP (SOAN-30), the composition of vegetation changed towards the predominance of pollen of herbaceous plants (94–95%) such as quinoa, wormwood, composites, grasses, and sedges. Of the tree species, there are single records of pollen grains of pine and birches [54]. The sequence of layers corresponds to the transition from the Karginsky interglacial to the Sartan stadial.

Near the mouth of the Suzun River, yet another section was studied. It presented the sediments of the first floodplain terrace of the Suzun river, taken from the side of river Ob and containing a large number of mollusk shells and plant remains. The obtained radiocarbon dates indicated the Late Glacial age of the peat layers (12,660 ± 130 BP (SOAN-1638) and 10,950 ± 150 BP (SOAN-54)) [54]. The spore-pollen spectra characterized landscapes of the forest-steppe type. Tree species were represented by pollen of spruce, pine, birch, and alder. Among the herbaceous pollens, mainly those of wormwood, quinoa, grasses, and several species of aquatic plants were present. The spore mosses were dominated by green and sphagnum mosses. Based on the weak presence of xerophytes, according to M.R. Votakh, the climatic conditions of 12,660 ± 130 BP were quite humid. Later, in 10,950 ± 150 BP, the conditions changed towards forest-tundra landscapes. The climate was reconstructed as being much drier and colder, with the expansion of the open, treeless landscapes [54].

A similar landscape composition can be found in the sediments on the Chulym River, where V.S. Volkova and I.A. Volkov [55] worked in the 1970s. They obtained palynological data from a lens of alluvial peat, which has no date but is the basal layer underlying deposits with a radiocarbon date of 21,800 BP. The palynological material there characterizes a landscape and climatic conditions close to the modern conditions, with the development of forest vegetation. However, the spore-pollen spectra of sediments dating back to MIS 2 (21,800 BP) were distinguished by the predominance of grass pollen (up to 50%), mainly haze and forbs. At the same time the content of tree pollen ranged from 30 to 25%. The palynological spectra given by V.S. Volkova and I.A. Volkov reflected the development of vegetation characteristic for the open swampy areas with meadows along rivers and lake shores. In addition, spores of tundra vegetation were found, which indicated the development of periglacial steppe-tundra vegetation [55].

Based on the abovementioned findings, it can be seen that, at the end of MIS 3 and even in MIS 2, tree species were present in the southern part of the West Siberian Plain, despite the absence of forest species of large and small mammals. However, the question of whether it was a forest zone or whether the trees spread only along the floodplains of the rivers remains open for further discussions. It is also a question whether the floodplain forest areas could have been refugia during the cold stages of the Pleistocene, which later formed the modern forest zone of the West Siberian Plain. The Suzun-1 and Suzun-2 entomocomplexes confirmed the distribution of open tundra-steppe landscapes at the beginning of MIS 2 in the south of the West Siberian Plain. At the same time, the presence in the entomological complexes of a large number of species characteristic of forest landscapes and their complete absence in other entomological complexes of the region indicate that during the cold periods of the Pleistocene, including the beginning of MIS 2, forests were not widespread in the south of the West Siberian Plain. The pollen of tree species in the palynological data as well as the remains of insects and crustaceans in our samples are manifestations of small areas of forest confined to floodplains and lakes remaining from

old river courses. For a small number of species, such depleted forest associations could be the centers of the formation of the modern forest entomofauna of the West Siberian Plain.

## 5. Conclusions

In the deposits of Suzun-1, Suzun-2, and Nizhny Suzun in the Upper Ob region, a complex of species of the arthropod orders Coleoptera, Hemiptera, Hymenoptera, Diptera (Insecta), Araneae (Arachnida), Daphniidae, and Triopsidae (Crustacea: Branchiopoda) were found. Diverse beetles, including 194 species from 21 families, were represented in all three deposits. In terms of species composition as well as the ecological preferences and distribution patterns of particular species, the studied entomocomplexes were consistent with the unique "*Otiorhynchus*-type" fauna that inhabited the southern part of the West Siberian Plain at the end of the Pleistocene. The cold and dry climate and prevailing open landscapes of the tundra-steppe type were the ecological conditions reconstructed for that fauna.

A distinctive feature of the Suzun-2 and Suzun-1 entomocomplexes is the relatively high proportion of forest species associated with both coniferous and deciduous (Suzun-1) species. According to these data, spruce forests with the inclusion of small-leaved species (birch) were being reconstructed at the beginning of MIS 2 in the Novosibirsk Ob region. The forest areas were probably isolated, confined to river valleys, and not widespread.

**Author Contributions:** The original draft was completed by A.A.G. and all coauthors. A.A.G., R.Y.D. and A.A.L. designed the study, suggested ideas for discussion, and prepared figures. The section descriptions were prepared by E.V.Z.; the insect fauna was studied by R.Y.D., E.V.Z. and A.A.G. (Coleoptera), A.V.I. (Scarabaeidae and Tenebrionidae), Y.E.M. (Chrysomelidae), A.A.P. (aquatic Coleoptera), A.S.P. (Elateridae), A.Y.S. (Staphylinidae) and A.A.L. (Brentidae, Curculionidae, and Scolytidae); the branchiopod crustaceans were studied by A.A.K. All authors contributed to the writing, discussion, reviewing, and editing of the manuscript. All authors have read and agreed to the published version of the manuscript.

**Funding:** The studies of A.G., R.D. and A.L. were carried out under the Federal Fundamental Scientific Research Program (grant No. 1021051703269-9-1.6.12). The studies of A.A.P. were carried out under the framework of Russian state research project No. 121051100109-1. Overall, this study was carried out under a government contract with the Institute of Plant and Animal Ecology, Ural Branch of the Russian Academy of Sciences (research subject: FUWU-2022-0005).

**Institutional Review Board Statement:** Not acceptable.

**Data Availability Statement:** Material was deposited in the collection of the Institute of Systematics and Ecology of Animals.

**Acknowledgments:** The authors are sincerely grateful to M.A. Kulkova (A. Herzen Russian State Pedagogical University, St. Petersburg) for determining the radiocarbon age, A.S. Sazhnev (Papanin Institute for Biology of Inland Waters, Russian Academy of Sciences, Borok) for the identification of Heteroceridae, and E.R. Dudko and M.S. Kireev (Novosibirsk) for help in collecting material. All SEM work was carried out at the Joint Service Center "Instrumental Methods in Ecology" (A.N. Severtsov Institute of Ecology and Evolution of the Russian Academy of Sciences). Many thanks to A.N. Neretina for her assistance with the SEM work.

**Conflicts of Interest:** The authors declare no conflict of interest. The funders had no role in the design of the study; in the collection, analyses, or interpretation of the data; in the writing of the manuscript; or in the decision to publish the results.

# Appendix A

**Table A1.** Subfossil insects from the Suzun-1, Suzun-2, and Nizhny Suzun sites.

| No | Species | N | | | | Nmin | | | | | | | | | | | Σ |
| | | | | | | Suzun-1 | | | | | Suzun-2 | | | | Nizhny Suzun | | |
| | | H | P | E | O | S1 | S2 | S3 | S4 | S5 | S1 | S2 | S3 | S4 | S1 | S2 | |
|---|---|---|---|---|---|---|---|---|---|---|---|---|---|---|---|---|---|
| | | | | | | DYTISCIDAE | | | | | | | | | | | |
| 1 | * *Agabus ?adpressus* Aube, 1837 | – | – | 1 | – | – | – | – | 1 | – | – | – | – | – | – | – | 1 |
| 2 | * *Agabus congener* (Thunberg, 1794) | – | 2 | – | – | – | – | 1 | – | 1 | – | – | – | – | – | – | 2 |
| 3 | * *Agabus coxalis* Sharp, 1882 | – | 1 | – | – | 1 | – | – | – | – | – | – | – | – | – | – | 1 |
| 4 | * *Agabus labiatus* (Brahm, 1791) | – | 1 | – | – | – | – | – | – | – | 1 | – | – | – | – | – | 1 |
| 5 | * *Agabus pallens* Poppius, 1905 | – | 5 | – | – | 2 | – | – | – | 1 | 1 | – | 1 | – | – | – | 5 |
| 6 | * *Ilybius subaeneus* Erichson, 1837 | – | 1 | – | 1 | – | – | – | 1 | 1 | – | – | – | – | – | – | 2 |
| 7 | *Ilybius* sp.1 | – | – | 1 | – | – | – | – | – | – | – | 1 | – | – | – | – | 1 |
| 8 | * *Nebrioporus ?depressus* (Fabricius, 1775) | – | 1 | – | – | 1 | – | – | – | – | – | – | – | – | – | – | 1 |
| 9 | * *Porhydrus lineatus* (Fabricius, 1775) | – | 1 | – | – | – | – | – | – | 1 | – | – | – | – | – | – | 1 |
| – | Dytiscidae indet. | – | 1 | – | – | – | – | – | – | – | – | – | 1 | – | – | – | 1 |
| | | | | | | CARABIDAE | | | | | | | | | | | |
| 10 | *Pelophila borealis* (Paykull, 1790) | – | 1 | 2 | – | 1 | – | – | 1 | – | – | – | 1 | – | – | – | 3 |
| 11 | *Notiophilus aquaticus/N.* cf. *aquaticus* (Linnaeus, 1758) | – | – | 4 | – | – | 1 | – | – | 1 | – | – | 1 | – | – | – | 3 |
| 12 | *Nebria gyllenhali* (Schönherr, 1806) | – | 2 | – | – | – | – | – | – | 1 | – | – | 1 | – | – | – | 2 |
| 13 | * *Carabus henningi* Fischer von Waldheim, 1817 | – | – | 1 | – | – | – | 1 | – | – | – | – | – | – | – | – | 1 |
| 14 | * *Carabus regalis* Fischer von Waldheim, 1820 | – | – | 1 | – | – | – | – | – | – | 1 | – | – | – | – | – | 1 |
| 15 | * *Carabus arvensis* Herbst, 1784 | – | – | 1 | – | – | – | – | – | – | – | – | – | – | 1 | – | 1 |
| 16 | *Diacheila polita* (Faldermann, 1835) | – | – | 1 | – | 1 | – | – | – | – | – | – | – | – | – | – | 1 |
| 17 | *Blethisa multipunctata* (Linnaeus, 1758) | – | – | 2 | – | 1 | 1 | – | – | – | – | – | – | – | – | – | 2 |
| 18 | *Clivina fossor* (Linnaeus, 1758) | – | 1 | 5 | – | – | – | 1 | 1 | 1 | – | 1 | 2 | – | – | – | 6 |
| 19 | * *Dyschiriodes* cf. *rufipes* (Dejean, 1825) | – | – | 1 | – | 1 | – | – | – | – | – | – | – | – | – | – | 1 |
| 20 | *Dyschiriodes tristis/D.* cf. *tristis* (Stephens, 1827) | – | 1 | 1 | – | – | – | – | 1 | – | – | – | – | – | – | – | 1 |
| – | *Dyschiriodes* sp. | – | – | 1 | – | – | – | – | 1 | – | – | – | – | – | – | – | 1 |
| 21 | * *Bembidion* (*Bracteon*) *lapponicum* Zetterstedt, 1828 | – | – | 1 | – | – | – | – | – | – | – | – | – | 1 | – | – | 1 |
| 22 | *Bembidion* (*Chlorodium*) *almum almum* J.Sahlberg, 1900 | – | 4 | 9 | – | 1 | 1 | 1 | 1 | 1 | – | – | 2 | 2 | – | 1 | 10 |
| 23 | * *Bembidion* (*Notaphus*) *obliquum/B.* cf. *obliquum* Sturm, 1825 | – | 2 | 6 | – | 1 | – | – | 1 | – | – | – | 2 | – | – | – | 4 |
| 24 | *Bembidion* (*Eupetedromus*) sp. | – | – | 1 | – | 1 | – | – | – | – | – | – | – | – | – | – | 1 |
| 25 | *Bembidion* (*Semicampa*) sp.1 | – | – | 2 | – | – | – | – | – | – | 1 | – | 1 | – | – | – | 2 |
| 26 | *Bembidion* (*Philochtus*) cf. *aeneum* Germar, 1823 | – | – | 2 | – | – | – | – | – | – | – | – | 1 | 1 | – | – | 2 |
| 27 | * *Bembidion* (*Bembidion*) cf. *paediscum* Bates, 1883 | – | 1 | 3 | – | 1 | – | 1 | – | – | – | – | 1 | – | – | – | 3 |
| 28 | *Bembidion* (*Plataphus*) *difficile* (Motschulsky, 1844) | – | – | 4 | – | 1 | 1 | 1 | – | – | – | – | 1 | – | – | – | 4 |
| 29 | *Bembidion* (*Ocydromus*) cf. *scopulinum* (Kirby 1837) | – | 1 | 2 | – | 1 | – | – | – | 1 | – | – | – | – | – | – | 2 |
| 30 | *Bembidion* (*Asioperyphus*) cf. *infuscatum* Dejean, 1831 | – | 2 | 5 | – | – | – | 1 | 1 | 1 | 1 | 1 | 1 | 1 | – | – | 7 |
| 31 | *Bembidion* (*Asioperyphus*) sp.1 | – | – | 7 | – | – | – | – | – | – | 1 | 1 | 2 | – | 1 | – | 5 |

Table A1. *Cont.*

| No | Species | N | | | | Nmin | | | | | | | | | | | Σ |
|----|---------|---|---|---|---|------|---|---|---|---|---|---|---|---|---|---|---|
| | | | | | | Suzun-1 | | | | | Suzun-2 | | | | Nizhny Suzun | | |
| | | H | P | E | O | S1 | S2 | S3 | S4 | S5 | S1 | S2 | S3 | S4 | S1 | S2 | |
| 32 | *Bembidion* (*Peryphus*) cf. *dauricum* (Motschulsky, 1844) | – | – | 3 | – | – | – | – | – | 1 | – | – | 1 | 1 | – | – | 3 |
| 33 | * *Bembidion* (*Peryphus*) cf. *jedlickai* Fassati, 1945 | – | – | 2 | – | – | 1 | – | – | – | – | – | – | – | – | – | 1 |
| 34 | *Bembidion* (*Peryphus*) *obscurellum* (Motschulsky, 1845) | – | – | 8 | – | – | – | – | 1 | – | 1 | 2 | 3 | 1 | – | – | 8 |
| 35 | *Bembidion* (*Testediolum*) *kokandicum* Solsky, 1874 | – | – | 5 | – | – | – | – | 1 | – | – | 1 | 1 | – | 1 | – | 4 |
| 36 | * *Bembidion* (*Pamirium*) cf. *roborovskii* Mikhailov, 1988 | – | 4 | 19 | – | 3 | – | 2 | 1 | 1 | 3 | – | 6 | 2 | – | – | 18 |
| – | *Bembidion* (*Ocydromus*) s.l.) spp. | – | 3 | – | – | 1 | – | – | 2 | – | – | – | – | – | – | – | 3 |
| – | *Bembidion* spp. | 4 | 8 | 5 | – | 3 | – | 1 | 1 | 2 | 1 | 1 | 3 | – | – | 1 | 13 |
| 37 | *Pogonus punctulatus* Dejean, 1828 | – | 2 | 4 | – | – | 1 | – | 1 | 1 | – | – | 1 | – | – | – | 4 |
| 38 | *Patrobus* cf. *septentrionis* Dejean, 1828 | 1 | 2 | 2 | – | – | – | – | – | 1 | 1 | 1 | 1 | – | 1 | – | 5 |
| 39 | *Poecilus* cf. *ravus* (Lutschnik, 1922) | – | 6 | 19 | – | 1 | 2 | 4 | 1 | 2 | – | 1 | 1 | 2 | 1 | – | 15 |
| 40 | *Pterostichus* (*Pseudomaseus*) *nigrita* (Paykull, 1790) | – | – | 1 | – | – | – | – | – | 1 | – | – | – | – | – | – | 1 |
| 41 | *Pterostichus* (*Phonias*) sp. | – | 2 | 4 | – | 1 | – | 1 | 1 | – | – | – | 2 | – | – | – | 5 |
| 42 | *Pterostichus* (*Cryobius*) sp.1 | – | – | 3 | – | 1 | – | 1 | – | – | – | – | – | – | – | – | 2 |
| 43 | *Pterostichus* (*Cryobius*) sp.2 | – | – | 1 | – | – | – | – | – | – | – | – | 1 | – | – | – | 1 |
| – | *Pterostichus* (*Cryobius*) spp. | – | 2 | 1 | – | – | 1 | – | 1 | – | – | – | 1 | – | – | – | 3 |
| 44 | *Pterostichus* (*Eosteropus*) cf. *maurusiacus* (Mannerheim, 1825) | – | – | 1 | – | – | 1 | – | – | – | – | – | – | – | – | – | 1 |
| 45 | * *Pterostichus* (*Petrophilus*) cf. *altainus* Jedlička, 1958 | – | – | 1 | – | – | 1 | – | – | – | – | – | – | – | – | – | 1 |
| – | *Pterostichus* spp. | – | – | 5 | – | – | 1 | 1 | 1 | 1 | – | – | 1 | – | – | – | 5 |
| 46 | * *Amara* (*Amara*) cf. *depressangula* Poppius, 1908 | – | – | 2 | – | 1 | – | – | – | – | 1 | – | – | – | – | – | 2 |
| 47 | * *Amara* (*Celia*) cf. *infima* (Duftschmid, 1812) | – | 1 | – | – | – | – | – | – | – | – | – | 1 | – | – | – | 1 |
| 48 | * *Amara* (*Celia*) *rupicola*/*A.* cf. *rupicola* Zimmermann, 1832 | – | 2 | 1 | – | – | – | 1 | 1 | – | – | – | – | – | – | – | 2 |
| 49 | * *Amara* (*Celia*) cf. *saginata* Ménétriés, 1847 | – | – | 1 | – | – | – | – | – | – | – | – | – | – | 1 | – | 1 |
| 50 | *Amara* (*Paracelia*) *quenseli* (Schönherr 1806) | – | 1 | – | – | – | – | – | – | – | – | – | – | – | – | 1 | 1 |
| 51 | * *Amara* (*Amathitis*) cf. *microdera* (Chaudoir, 1844) | – | 1 | – | – | – | – | – | – | 1 | – | – | – | – | – | – | 1 |
| 52 | *Curtonotus* cf. *alpinus* (Paykull, 1790) | – | 1 | – | – | – | – | – | – | 1 | – | – | – | – | – | – | 1 |
| 53 | * *Curtonotus* cf. *fodinae* (Mannerheim, 1825) | – | 1 | 1 | – | 1 | – | – | – | – | – | – | 1 | – | – | – | 2 |
| – | *Curtonotus* sp. | – | – | 1 | – | – | – | – | – | – | – | – | – | – | 1 | – | 1 |
| 54 | * *Agonum carbonarium*/*A.* cf. *carbonarium* Dejean, 1828 | – | 1 | 2 | – | – | 1 | – | – | – | – | 1 | – | – | – | – | 2 |
| – | *Agonum* sp. | – | – | 3 | – | – | – | – | 1 | 1 | 1 | – | – | – | – | – | 3 |
| 55 | *Dicheirotrichus* (*Trichocellus*) sp. | – | – | 4 | – | – | – | 1 | – | – | 1 | 1 | 1 | – | – | – | 4 |
| 56 | *Harpalus amputatus* Say, 1830 | – | 8 | 9 | – | 1 | 1 | – | 1 | 2 | 1 | 2 | – | 2 | – | – | 10 |
| 57 | * *Harpalus anxius*-group (Duftschmid, 1812) | – | – | 3 | – | – | 1 | – | – | – | – | – | – | – | – | – | 1 |
| 58 | * *Harpalus pusillus*-group (Motschulsky, 1850) | – | 1 | – | – | – | – | – | – | – | 1 | – | – | – | – | – | 1 |
| 59 | * *Harpalus salinus* Dejean, 1829 | – | – | 2 | – | 1 | – | – | – | – | – | – | – | 1 | – | – | 2 |

**Table A1.** *Cont.*

| No | Species | N | | | | Nmin Suzun-1 | | | | | Suzun-2 | | | | Nizhny Suzun | | Σ |
|----|---------|---|---|---|---|----|----|----|----|----|----|----|----|----|----|----|---|
|    |         | H | P | E | O | S1 | S2 | S3 | S4 | S5 | S1 | S2 | S3 | S4 | S1 | S2 |   |
| 60 | *Harpalus* sp.1 | – | – | 3 | – | – | – | – | 1 | 1 | – | – | – | – | – | – | 2 |
| – | *Harpalus* spp. | – | – | 4 | – | 2 | – | – | – | – | – | – | – | 1 | – | – | 3 |
| 61 | *Cymindis binotata/C.* cf. *binotata* Fischer von Waldheim, 1820 | – | – | 4 | – | 2 | – | 1 | – | – | – | – | 1 | – | – | – | 4 |
| 62 | *Cymindis* cf. *kasakh* Kryzhanovskij et Emetz, 1973 | – | 1 | 1 | – | – | – | – | – | – | 1 | – | 1 | – | – | – | 2 |
| 63 | *Syntomus* sp. | – | – | 2 | – | – | 1 | – | – | – | – | – | 1 | – | – | – | 2 |
| 64 | *Lebia punctata* Gebler, 1843 | – | – | 1 | – | – | 1 | – | – | – | – | – | – | – | – | – | 1 |
| – | Carabidae indet. | 3 | 3 | 8 | – | 1 | – | – | 1 | 2 | – | – | – | 1 | – | – | 5 |
| | HELOPHORIDAE | | | | | | | | | | | | | | | | |
| 65 | *Helophorus obscurellus* Poppius, 1907 | – | 2 | 3 | – | 1 | – | 1 | 1 | 1 | – | – | 1 | – | – | – | 5 |
| 66 | * *Helophorus orientalis* Motschulsky, 1860 | – | 1 | 1 | – | 1 | – | – | – | – | – | – | – | – | – | – | 1 |
| 67 | * *Helophorus pallidus* Gebler, 1830 | – | 1 | 1 | – | – | – | – | – | – | – | – | 1 | – | – | – | 1 |
| 68 | * *Helophorus ?parajacutus* Angus, 1970 | – | – | 1 | – | 1 | – | – | – | – | – | – | – | – | – | – | 1 |
| – | *Helophorus* sp. | – | – | 2 | – | – | – | – | – | 1 | – | – | 2 | – | – | – | 3 |
| | HYDROPHILIDAE | | | | | | | | | | | | | | | | |
| 69 | *Enochrus quadripunctatus* (Herbst, 1797) | – | – | – | 1 | – | – | – | – | – | 1 | – | – | – | – | – | 1 |
| 70 | *Hydrobius* sp. | – | – | 1 | – | – | – | – | – | – | 1 | – | – | – | – | – | 1 |
| – | Hydrophilidae indet. | – | – | 2 | – | – | – | – | – | – | – | – | – | 1 | – | – | 1 |
| | HYDRAENIDAE | | | | | | | | | | | | | | | | |
| 71 | *Ochthebius* (?*Asiobates*) sp.1 | – | – | 3 | – | 1 | – | – | – | – | – | – | 2 | – | – | – | 3 |
| 72 | *Ochthebius* sp.2 | – | – | 1 | – | – | – | – | 1 | – | – | – | – | – | – | – | 1 |
| | LEIODIDAE | | | | | | | | | | | | | | | | |
| 73 | Leiodidae indet. | – | – | 6 | – | 2 | – | – | – | 1 | – | – | 2 | – | – | – | 5 |
| | SILPHIDAE | | | | | | | | | | | | | | | | |
| 74 | *Aclypea bicarinata* (Gebler, 1830) | – | – | 1 | – | – | – | – | – | 1 | – | – | – | – | – | – | 1 |
| 75 | *Aclypea calva* (Reitter, 1890) | – | 1 | – | – | – | – | – | – | 1 | – | – | – | – | – | – | 1 |
| 76 | *Aclypea opaca* (Linnaeus, 1758) | – | 4 | 8 | – | 2 | 1 | 1 | – | 1 | – | – | 1 | 2 | – | 1 | 9 |
| 77 | *Aclypea sericea* (Zoubkoff, 1833) | – | – | 7 | – | 1 | – | 1 | – | – | 1 | – | 1 | 1 | – | – | 5 |
| – | *Aclypaea* sp. | – | 1 | – | – | – | 1 | – | – | – | – | – | – | – | – | – | 1 |
| 78 | * *Silpha carinata* Herbst, 1783 | – | 1 | – | – | – | 1 | – | – | – | – | – | – | – | – | – | 1 |
| 79 | *Thanatophilus trituberculatus* (Kirby, 1837) | – | 1 | – | – | – | – | – | – | – | 1 | – | – | – | – | – | 1 |
| – | Silphidae indet. | – | 17 | 2 | 1 | 1 | 2 | 1 | 1 | 2 | 1 | – | 2 | 4 | – | – | 14 |
| | STAPHYLINIDAE | | | | | | | | | | | | | | | | |
| 80 | Aleocharinae indet. sp.1 | – | 1 | – | – | 1 | – | – | – | – | – | – | – | – | – | – | 1 |
| 81 | Aleocharinae indet. sp.2 | – | 5 | – | – | 3 | – | 1 | – | – | – | – | 1 | – | – | – | 5 |
| 82 | Aleocharinae indet. sp.3 | – | 2 | – | – | – | – | 1 | – | – | – | – | 1 | – | – | – | 2 |
| 83 | Aleocharinae indet. sp.4 | – | 5 | – | – | 3 | – | – | 2 | – | – | – | – | – | – | – | 5 |
| 84 | Aleocharinae indet. sp.5 | – | 1 | – | – | – | – | 1 | – | – | – | – | – | – | – | – | 1 |
| 85 | Aleocharinae indet. sp.6 | – | 1 | – | – | – | – | – | – | – | 1 | – | – | – | – | – | 1 |
| 86 | Aleocharinae indet. sp.7 | – | 1 | – | – | – | – | – | – | – | – | 1 | – | – | – | – | 1 |
| 87 | Aleocharinae indet. sp.8 | – | 1 | – | – | – | – | – | – | – | – | – | – | 1 | – | – | 1 |
| 88 | Aleocharinae indet. sp.9 | – | 1 | – | – | – | – | 1 | – | – | – | – | – | – | – | – | 1 |
| 89 | Aleocharinae indet. sp.10 | – | 1 | – | – | – | – | 1 | – | – | – | – | – | – | – | – | 1 |
| – | Aleocharinae indet. spp. | – | – | 1 | – | 1 | – | – | – | – | – | – | – | – | – | – | 1 |

**Table A1.** *Cont.*

| No | Species | N | | | | Nmin | | | | | | | | | | | Σ |
|----|---------|---|---|---|---|---|---|---|---|---|---|---|---|---|---|---|---|
| | | | | | | Suzun-1 | | | | | Suzun-2 | | | | Nizhny Suzun | | |
| | | H | P | E | O | S1 | S2 | S3 | S4 | S5 | S1 | S2 | S3 | S4 | S1 | S2 | |
| 90 | * *Olophrum* cf. *fuscum*(Gravenhorst, 1806) | – | 2 | 10 | – | 2 | – | – | 1 | 1 | 2 | – | 2 | – | – | – | 8 |
| 91 | *Olophrum* sp.1 | – | 1 | 6 | – | 1 | – | 1 | – | 1 | 1 | – | 1 | – | – | – | 5 |
| 92 | * *Phloeostiba lapponica* (Zetterstedt, 1838) | – | 1 | – | – | – | – | – | 1 | – | – | – | – | – | – | – | 1 |
| – | Omaliinae sp. | – | 1 | 1 | – | 1 | – | – | – | – | – | – | – | – | – | – | 1 |
| 93 | *Bledius* sp. | – | 6 | 7 | 1 | 2 | – | 2 | 3 | 1 | – | – | – | – | – | – | 8 |
| 94 | * *Platystethus* cf. *cornutus* (Gravenhorst, 1802) | 7 | 13 | 7 | – | 3 | – | 4 | 2 | 5 | 2 | 1 | 1 | – | – | – | 18 |
| 95 | *Lathrobium* sp. | – | 2 | – | – | 1 | – | – | 1 | – | – | – | – | – | – | – | 2 |
| 96 | *Ochthephilum* sp. | – | – | 2 | – | – | – | – | – | – | 1 | – | – | – | – | – | 1 |
| 97 | *Philonthus* sp.1 | – | – | 1 | – | 1 | – | – | – | – | – | – | – | – | – | – | 1 |
| 98 | *Philonthus* sp.2 | – | – | 2 | – | – | – | – | – | – | 1 | – | 1 | – | – | – | 2 |
| 99 | *Philonthus* sp.3 | – | – | 1 | – | – | – | – | – | 1 | – | – | – | – | – | – | 1 |
| 100 | *Philonthus* sp.4 | – | 1 | – | – | – | – | – | 1 | – | – | – | – | – | – | – | 1 |
| 101 | *Stenus* sp. | – | – | 1 | – | – | – | 1 | – | – | – | – | – | – | – | – | 1 |
| 102 | *Lordithon* sp. | – | – | 2 | – | 1 | – | – | 1 | – | – | – | – | – | – | – | 2 |
| 103 | *Mycetoporus* sp. | – | 1 | – | – | – | – | – | – | – | – | – | 1 | – | – | – | 1 |
| 104 | * *Tachinus* cf. *rufipes* Linnaeus,1758 | – | – | 1 | – | – | – | – | – | – | 1 | – | – | – | – | – | 1 |
| 105 | *Tachyporus* spp. | – | 1 | 1 | – | 1 | – | – | – | – | – | – | 1 | – | – | – | 2 |
| 106 | Staphylinidae indet. sp.1 | – | – | 1 | – | – | – | 1 | – | – | – | – | – | – | – | – | 1 |
| – | Staphylinidae indet. | – | 2 | – | 1 | 1 | – | 1 | – | – | – | – | 1 | – | – | – | 3 |
| | SCARABAEIDAE | | | | | | | | | | | | | | | | |
| 107 | * *Aphodius* (*Acanthobodilus*) cf. *immundus* Creutzer, 1799 | – | – | 1 | – | – | – | – | – | – | – | – | 1 | – | – | – | 1 |
| 108 | * *Aphodius* (*Bodilus*) cf. *lugens* (Creutzer, 1799) | – | – | 2 | – | 1 | – | – | – | – | 1 | – | – | – | – | – | 2 |
| 109 | *Aphodius* (*Chilothorax*) *melanostictus* W.Schmidt, 1840 | – | – | 1 | – | – | – | – | – | – | – | 1 | – | – | – | – | 1 |
| 110 | * *Aphodius* (*Colobopterus*) *erraticus* (Linnaeus, 1758) | – | – | 1 | – | – | – | – | 1 | – | – | – | – | – | – | – | 1 |
| 111 | *Aphodius* (*Liothorax*) *plagiatus*/*A.* cf. *plagiatus* (Linnaeus, 1767) | – | – | 7 | – | – | 1 | – | 2 | 2 | – | – | – | 1 | – | – | 6 |
| 112 | * *Aphodius* (*Loraspis*) *frater* Mulsant et Rey, 1870 | – | – | 1 | – | – | – | – | – | – | – | – | – | – | 1 | – | 1 |
| 113 | *Aphodius* (*Nobius*) cf. *serotinus* (Panzer, 1799) | – | – | 13 | – | 2 | 1 | 2 | – | – | – | 2 | 3 | 1 | – | – | 11 |
| 114 | *Aphodius* (*Phaeaphodius*) *rectus* Motschulsky, 1866 | – | – | 1 | – | – | – | – | – | 1 | – | – | – | – | – | – | 1 |
| 115 | * *Aphodius* (*Subrinus*) *sturmi* (Harold, 1870) | – | – | 1 | – | 1 | – | – | – | – | – | – | – | – | – | – | 1 |
| – | *Aphodius* spp. | 3 | 2 | 8 | – | 1 | – | 2 | 1 | 1 | – | – | 1 | 1 | 1 | – | 8 |
| 116 | *Aegialia* sp. | – | – | 2 | – | – | – | – | 1 | – | – | – | – | – | 1 | – | 2 |
| | BYRRHIDAE | | | | | | | | | | | | | | | | |
| 117 | *Porcinolus murinus* (Fabricius, 1794) | – | 1 | 4 | – | – | – | 1 | 1 | 2 | – | – | – | – | – | – | 4 |
| 118 | *Morychus ostasiaticus* Tshernyshev, 1997 | – | 5 | 12 | – | 1 | 1 | 1 | 2 | 3 | – | – | 1 | 1 | – | – | 10 |
| – | Byrrhidae indet. | – | – | 1 | – | – | – | – | – | – | – | – | – | – | – | 1 | 1 |
| | HETEROCERIDAE | | | | | | | | | | | | | | | | |
| 119 | *Augyles* sp. | – | – | – | 1 | – | – | – | – | – | – | 1 | – | – | – | – | 1 |
| 120 | * *Heterocerus marginatus* (Fabricius, 1787) | – | – | 1 | – | – | – | 1 | – | – | – | – | – | – | – | – | 1 |

**Table A1.** *Cont.*

| No | Species | N | | | | Nmin | | | | | | | | | | | Σ |
|---|---|---|---|---|---|---|---|---|---|---|---|---|---|---|---|---|---|
| | | | | | | Suzun-1 | | | | | Suzun-2 | | | | Nizhny Suzun | | |
| | | H | P | E | O | S1 | S2 | S3 | S4 | S5 | S1 | S2 | S3 | S4 | S1 | S2 | |
| | ELATERIDAE | | | | | | | | | | | | | | | | |
| 121 | *Berninelsonius hyperboreus* (Gyllenhal, 1827) | – | – | 1 | – | 1 | – | – | – | – | – | – | – | – | – | – | 1 |
| 122 | *Hypnoidus* cf. *rivularius* (Gyllenhal, 1808) | – | – | 3 | – | – | – | – | 1 | – | 1 | – | 1 | – | – | – | 3 |
| 123 | * *Hypoganomorphus laevicollis* (Mannerheim, 1852) | – | – | 2 | – | – | – | – | – | – | – | – | 1 | – | – | – | 1 |
| 124 | * *Pristilophus punctatissimus* (Ménétriés, 1851) | – | – | – | 1 | 1 | – | – | – | – | – | – | – | – | – | – | 1 |
| | BUPRESTIDAE | | | | | | | | | | | | | | | | |
| 125 | Buprestidae? indet. | – | 1 | – | – | – | – | – | – | – | – | – | 1 | – | – | – | 1 |
| | MALACHIIDAE | | | | | | | | | | | | | | | | |
| 126 | Malachiidae? indet. | – | – | 2 | – | 1 | – | – | – | – | – | – | 1 | – | – | – | 2 |
| | MELOIDAE | | | | | | | | | | | | | | | | |
| 127 | * *Mylabris ledebouri* Gebler, 1829 | – | – | 1 | – | – | – | – | – | – | – | – | 1 | – | – | – | 1 |
| – | *Mylabris* sp. | – | – | 1 | – | – | – | – | – | – | – | – | 1 | – | – | – | 1 |
| | TENEBRIONIDAE | | | | | | | | | | | | | | | | |
| 128 | *Centorus rufipes* (Gebler, 1833) | – | – | 13 | – | 2 | 1 | 1 | 2 | 1 | 1 | 1 | 1 | 1 | – | – | 11 |
| 129 | * *Centorus ?crassipes borealis* (Fischer von Waldheim, 1844) | – | – | 1 | – | 1 | – | – | – | – | – | – | – | – | – | – | 1 |
| – | *Centorus* spp. | 1 | 3 | 11 | 1 | 2 | 1 | 1 | 1 | 2 | 1 | – | 1 | 1 | – | – | 10 |
| 130 | *Platyscelis* sp. | 1 | – | – | 1 | – | – | 1 | 1 | – | – | – | – | – | – | – | 2 |
| 131 | *Scythis* sp. | 2 | 2 | – | – | – | – | – | 1 | 1 | – | – | – | – | – | – | 2 |
| 132 | Tenebrionidae indet. sp.1 | 1 | – | 2 | – | 1 | – | – | – | – | – | – | – | – | – | 1 | 2 |
| | CERAMBYCIDAE | | | | | | | | | | | | | | | | |
| 133 | * *Eodorcadion carinatum* (Fabricius, 1781) | – | 1 | – | – | – | 1 | – | – | – | – | – | – | – | – | – | 1 |
| | CHRYSOMELIDAE | | | | | | | | | | | | | | | | |
| 134 | * *Donacia dentata* Hoppe, 1795 | – | – | 2 | – | – | 1 | 1 | – | – | – | – | – | – | – | – | 2 |
| 135 | * *Plateumaris sericea* (Linnaeus, 1761) | – | – | 1 | – | – | – | 1 | – | – | – | – | – | – | – | – | 1 |
| 136 | * *Chrysolina* cf. *gebleri* L.Medvedev, 1979 | – | 1 | – | – | 1 | – | – | – | – | – | – | – | – | – | – | 1 |
| 137 | * *Chrysolina graminis artemisiae* (Motschulsky, 1860) | – | 1 | – | – | – | – | – | – | – | – | 1 | – | – | – | – | 1 |
| 138 | * *Crosita altaica* Gebler, 1823 | – | – | 1 | – | – | – | – | – | – | – | 1 | – | – | – | – | 1 |
| 139 | * *Colaphellus alpicola* (Warchałowski, 2004) | – | 3 | – | – | 1 | 1 | – | – | – | – | – | 1 | – | – | – | 3 |
| 140 | * *Charaea minutum* (Joannis, 1865) | – | – | 2 | – | – | – | – | – | – | – | 1 | 1 | – | – | – | 2 |
| 141 | * *Galeruca ?daurica* Joannis, 1866 | – | – | 1 | – | – | – | – | – | – | 1 | – | – | – | – | – | 1 |
| 142 | * *Luperus* cf. *longicornis* (Fabricius, 1781) | – | – | 2 | – | – | – | – | – | – | – | 1 | 1 | – | – | – | 2 |
| 143 | * *Hippuriphila modeeri* (Linnaeus, 1761) | – | – | 1 | – | – | – | – | – | 1 | – | – | – | – | – | – | 1 |
| 144 | * *Phyllotreta ?tetrastigma* (Comolli, 1837) | – | – | 1 | – | – | – | – | – | – | – | – | 1 | – | – | – | 1 |
| 145 | * *Phyllotreta nemorum* (Linnaeus, 1758) | – | – | 6 | – | – | 1 | 1 | – | – | – | 1 | 1 | 1 | – | – | 5 |
| – | Alticinae indet. | – | 2 | 1 | – | 1 | – | – | – | – | – | – | 1 | – | – | – | 2 |

**Table A1.** *Cont.*

| No | Species | N | | | | Nmin | | | | | | | | | | | Σ |
|---|---|---|---|---|---|---|---|---|---|---|---|---|---|---|---|---|---|
| | | | | | | Suzun-1 | | | | | Suzun-2 | | | | Nizhny Suzun | | |
| | | H | P | E | O | S1 | S2 | S3 | S4 | S5 | S1 | S2 | S3 | S4 | S1 | S2 | |
| 146 | * *Pachybrachis scriptidorsum* Marseul, 1875 | – | – | 1 | – | – | – | – | – | – | – | – | 1 | – | – | – | 1 |
| – | Chrysomelidae indet. | – | 4 | 36 | – | 1 | 3 | 1 | 2 | 2 | 3 | 1 | 3 | 1 | 1 | 2 | 20 |
| | BRENTIDAE | | | | | | | | | | | | | | | | |
| 147 | * *Taphrotopium ircutense* (Faust, 1888) | – | 1 | – | – | – | – | – | – | – | – | – | 1 | – | – | – | 1 |
| 148 | *Taphrotopium steveni* (Gyllenhal, 1839) | – | – | 1 | – | – | – | – | 1 | – | – | – | – | – | – | – | 1 |
| 149 | *Eutrichapion facetum* (Gyllenhal, 1839) | – | – | 1 | – | – | – | – | – | – | – | – | – | 1 | – | – | 1 |
| 150 | * *Ceratapion onopordi* (Kirby, 1808) | – | – | 1 | – | – | – | – | – | – | – | – | 1 | – | – | – | 1 |
| 151 | *Cyanapion* sp. | – | – | 1 | – | – | – | – | – | 1 | – | – | – | – | – | – | 1 |
| 152 | * *Ompholapion hookerorum* (Kirby, 1808) | – | – | 2 | – | – | – | – | – | 2 | – | – | – | – | – | – | 2 |
| 153 | *Trichapion simile* (Kirby, 1811) | – | – | 1 | – | – | – | – | 1 | – | – | – | – | – | – | – | 1 |
| – | Apioninae indet. | 1 | 1 | 1 | – | 1 | – | – | 1 | – | – | – | – | – | – | – | 2 |
| | CURCULIONIDAE | | | | | | | | | | | | | | | | |
| 154 | *Tournotaris bimaculata* (Fabricius, 1787) | 23 | 14 | 160 | – | 9 | 4 | 11 | 13 | 18 | 4 | 1 | 7 | 4 | – | 1 | 72 |
| 155 | * *Notaris scirpi* (Fabricius, 1792) | – | – | 1 | – | – | – | – | – | – | – | – | – | – | – | 1 | 1 |
| 156 | *Bagous* sp.1 | – | – | 2 | – | – | – | – | 1 | 1 | – | – | – | – | – | – | 2 |
| 157 | *Bagous* sp.2 | 1 | 1 | 2 | – | 1 | – | 1 | – | – | – | – | – | 1 | – | – | 3 |
| 158 | *Stephanocleonus foveifrons* Chevrolat, 1873 | 1 | – | 6 | 2 | 1 | 1 | – | – | 1 | 1 | 1 | – | 1 | – | – | 6 |
| 159 | *Stephanocleonus suvorovi* Legalov, 1999 | 1 | – | – | – | 1 | – | – | – | – | – | – | – | – | – | – | 1 |
| 160 | * *Baris artemisiae* (Herbst, 1795) | – | 1 | – | – | 1 | – | – | – | – | – | – | – | – | – | – | 1 |
| 161 | *Aulacobaris lepidii* (Germar, 1823) | 1 | 1 | 6 | – | 1 | – | 1 | – | 3 | – | – | 1 | – | – | – | 6 |
| 162 | * *Phytobius leucogaster* (Marsham, 1802) | – | 1 | 1 | – | – | – | – | – | – | 1 | – | – | 1 | – | – | 2 |
| 163 | *Pelenomus* sp. | – | – | 1 | – | – | – | – | 1 | – | – | – | – | – | – | – | 1 |
| 164 | *Ceutorhynchus* sp.1 | 1 | 1 | – | – | – | – | 1 | – | – | – | – | – | – | – | – | 1 |
| 165 | *Ceutorhynchus* sp.2 | – | – | 1 | – | – | – | – | – | – | – | – | 1 | – | – | – | 1 |
| 166 | * *Anoplus plantaris* (Næzén, 1794) | – | – | 1 | – | – | – | – | 1 | – | – | – | – | – | – | – | 1 |
| 167 | * *Tychius quinquepunctatus* (Linnaeus, 1758) | – | – | 1 | – | – | – | – | – | – | – | – | – | 1 | – | – | 1 |
| 168 | *Tychius* sp.1 | – | – | 1 | – | – | – | – | – | – | 1 | – | – | – | – | – | 1 |
| 169 | * *Eremochorus steppensis* (Motschulsky, 1860) | 2 | – | – | – | – | – | – | – | 2 | – | – | – | – | – | – | 2 |
| 170 | *Trichalophus biguttatus* Gebler, 1832 | 2 | – | – | – | – | – | – | – | 2 | – | – | – | – | – | – | 2 |
| 171 | * *Sitona obscuratus* Faust, 1882 | 1 | – | – | – | – | – | – | – | 1 | – | – | – | – | – | – | 1 |
| 172 | *Sitona ovipennis* Hochhuth, 1851 | – | – | 1 | – | – | – | – | 1 | – | – | – | – | – | – | – | 1 |
| 173 | *Sitona* sp.1 | – | 1 | – | – | – | – | – | 1 | – | – | – | – | – | – | – | 1 |
| 174 | *Sitona* sp.2 | – | – | 1 | – | 1 | – | – | – | – | – | – | – | – | – | – | 1 |
| 175 | *Sitona* sp. 3 | 1 | – | – | – | – | – | – | – | – | – | – | – | – | 1 | – | 1 |
| 176 | *Chlorophanus sibiricus* Gyllenhal, 1834 | – | – | 5 | – | 1 | – | – | – | 1 | 1 | – | – | – | – | – | 3 |
| 177 | *Eusomus ovulum* Germar, 1823 | – | – | 1 | – | – | – | – | – | – | – | – | 1 | – | – | – | 1 |
| 178 | *Paophilus albilaterus* (Faust, 1882) | – | – | 6 | – | – | – | – | 1 | – | 1 | 1 | 1 | – | – | – | 4 |
| 179 | * *Phyllobius crassipes* Motschulsky, 1860 | – | – | 1 | – | – | – | – | – | 1 | – | – | – | – | – | – | 1 |
| 180 | *Phyllobius pomaceus* Gyllenhal, 1834 | – | – | 1 | – | – | – | – | – | – | 1 | – | – | – | – | – | 1 |

Table A1. *Cont.*

| No | Species | N | | | | Nmin | | | | | | | | | | | | Σ |
|---|---|---|---|---|---|---|---|---|---|---|---|---|---|---|---|---|---|---|
| | | | | | | Suzun-1 | | | | | Suzun-2 | | | | Nizhny Suzun | | | |
| | | H | P | E | O | S1 | S2 | S3 | S4 | S5 | S1 | S2 | S3 | S4 | S1 | S2 | | |
| 181 | *Phyllobius virideaeris* (Laicharting, 1781) | 1 | 2 | 9 | – | 2 | – | – | 1 | – | 1 | 1 | 3 | – | – | – | 8 |
| 182 | *Phyllobius* sp.1 | – | – | 1 | – | – | – | – | – | – | – | – | – | – | – | 1 | 1 |
| 183 | *Phyllobius* sp.2 | – | – | 1 | – | – | – | – | – | – | – | – | – | – | – | 1 | 1 |
| 184 | *Polydrusus amoenus* (Germar, 1823) | – | – | 2 | – | – | – | 1 | – | – | – | – | 1 | – | – | – | 2 |
| 185 | * *Polydrusus corruscus* Germar, 1823 | – | – | 1 | – | – | – | – | 1 | – | – | – | – | – | – | – | 1 |
| 186 | * *Eudipnus mollis* (Stroem, 1768) | – | – | 1 | – | – | – | – | – | 1 | – | – | – | – | – | – | 1 |
| 187 | *Otiorhynchus politus* Gyllenhal, 1834 | 4 | 5 | – | – | 2 | – | 1 | 1 | 1 | 1 | – | – | 2 | – | – | 8 |
| 188 | *Otiorhynchus pullus* Gyllenhal, 1834 | 3 | 3 | 1 | – | – | – | 1 | 1 | 1 | – | 1 | – | 1 | – | 1 | 6 |
| 189 | *Otiorhynchus subocularis* L. Arnoldi, 1975 | 9 | 12 | 19 | – | 3 | 4 | – | 2 | 4 | 2 | 1 | 1 | 2 | – | – | 19 |
| 190 | *Otiorhynchus* af. *ursus* Gebler, 1844 | 17 | 28 | 44 | – | 4 | 3 | 3 | 6 | 6 | 2 | 1 | 4 | 4 | – | – | 33 |
| – | *Otiorhynchus* af. *ursus/O. bardus* Boheman, 1842 | 2 | – | – | – | – | – | – | – | – | – | – | – | – | – | 2 | 2 |
| 191 | *Otiorhynchus unctuosus* Germar, 1823 | – | – | 3 | – | – | – | – | – | – | – | – | – | – | – | 2 | 2 |
| 192 | *Otiorhynchus obscurus* Gyllenhal, 1834 | 1 | 2 | 4 | – | – | – | – | – | – | – | – | – | – | 1 | 2 | 3 |
| – | Curculionidae indet. | – | 11 | 17 | – | 3 | 1 | 1 | 3 | 2 | 2 | – | 3 | – | – | 1 | 16 |
| SCOLYTIDAE | | | | | | | | | | | | | | | | | |
| 193 | *Phloeotribus spinulosus* (Rey, 1883) | – | – | 26 | – | 3 | – | – | 5 | 5 | 1 | – | 2 | – | – | – | 18 |
| 194 | * *Polygraphus subopacus* Thomson, 1871 | – | – | 1 | – | – | – | – | 1 | – | – | – | – | – | – | – | 1 |
| – | COLEOPTERA indet. | 5 | 9 | 28 | 5 | 4 | – | 2 | 3 | 5 | 1 | – | 2 | – | – | – | 17 |
| **Coleoptera in total** | | 100 | 277 | 795 | 16 | 123 | 43 | 78 | 105 | 123 | 57 | 33 | 125 | 49 | 11 | 21 | 770 |
| **Number of Coleoptera species** | | 25 | 83 194 | 144 | 10 | 67 | 27 | 45 145 | 59 | 60 | 39 | 28 101 | 74 | 27 | 11 23 | 16 | 194 |
| HEMIPTERA indet. | | – | 3 | 1 | – | 1 | – | – | 1 | – | 1 | – | 1 | – | – | – | 4 |
| HYMENOPTERA indet. | | 4 | 1 | – | 2 | 2 | 1 | 1 | 1 | 1 | – | – | – | – | – | – | 6 |
| DIPTERA indet. | | – | – | – | 1 | – | – | – | 1 | – | – | – | – | – | – | – | 1 |
| INSECTA indet. | | 2 | – | 1 | 7 | 1 | – | – | 1 | 1 | 1 | – | – | – | – | – | 4 |
| ARANEI indet. | | – | – | – | 2 | – | – | 1 | 1 | – | – | – | – | – | – | – | 2 |

Notes: N—number of fragments: H—head, P—pronotum, E—elytron, O—other fragments. Nmin—minimum number of individuals. S1–S5—samples. * species records for Pleistocene deposit of West Siberia for the first time.

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
