# Peer review of "New Data on the Distribution of Southern Forests for the West Siberian Plain during the Late Pleistocene: A Paleoentomological Approach"

_diversity, doi:10.3390/d15010056_

Round 1
Reviewer 1 Report
I have reviewed the manuscript Diversity-2129276, “Beetles and branchiopod crustaceans from the Quaternary deposits in the lower current of Suzun River (Upper Ob River basin): new data about distribution of forests in the south of West Siberian Plain in the late Pleistocene” and submitted my review, revisions, and comments on December 17, 2022.
In this manuscript, they documented “Subfossil remains of insects and branchiopod crustaceans (Cladocera and Notostraca) found in three late Pleistocene deposits in the Novosibirsk region in the vicinity of the village of Suzun” and presented that “At least 194 beetle species from 21 families have been found altogether, of them, 74 species were found in the Pleistocene deposits of Western Siberia for the first time. All deposits are similar in species composition of beetles; Carabidae and Curculionidae prevail everywhere. The ecological composition is dominated by steppe and tundra-steppe species; aquatic and riparian groups are also well represented. According to these data, at the beginning of MIS 2 in the Upper Ob region, spruce forests with the participation of small-leaved species (birch) were present. Probably they were confined to river valleys and were not widely distributed.”
The co-authors have done a good job collecting these fossil specimens, analyzing morphological characters of the subfossil remains or fragments and studying the ecological compositions, coexisting plants and/or other animals. They have provided clear and detailed data and comparisons of the new specimens with figures, photos and Tables.
Here are some key revisions and changes:
· The title is too long. Suggest deleting the second part as marked by yellow highlight.
· This term of “Quaternary” is used in the title. Delete this word from the list of keywords because all title words are treated as keywords.
· MIS 3 is “Marine isotope stage 3”, not “Marine isotope state 3”.
· The scale bar for Figure 3 f is missing.
· In the Figure 4, two scale bars are marked as “2 mm for Figs 1, 4, 5 and for Figs 2. 3”. Please changed the numbers to a, b, c, d, and e.
· In the section of Discussion 4.1, the last sentence of the second paragraph is not complete. Please complete this sentence!
· For Figure 6, please clarify the meaning of the horizontal scales.
In the attached PDF version of the manuscript, I used the Open Comments and yellow highlights to indicate suggested revisions and improvements for this paper.

Author Response
Dear Reviewer, We express our sincere gratitude to you for reviewing our work and making very valuable comments. We have taken your comments into account and made the appropriate corrections.
- The title is too long. Suggest deleting the second part as marked by yellow highlight.- the Title has been changed.
- The This term of “Quaternary” is used in the title. Delete this word from the list of keywords because all title words are treated as keywords. - Due to the fact that the Title has changed, we did not remove the word "Quaternary" from keywords.
- MIS 3 is “Marine isotope stage 3”, not “Marine isotope state 3” - It was done.
- The scale bar for Figure 3 f is missing.- It was done.
- In the Figure 4, two scale bars are marked as “2 mm for Figs 1, 4, 5 and for Figs 2. 3”. Please changed the numbers to a, b, c, d, and e.- It was done.
- In the section of Discussion 4.1, the last sentence of the second paragraph is not complete. Please complete this sentence! - It was done.
- For Figure 6, please clarify the meaning of the horizontal scales. - It was done.
Wit best wishes, Anna Gurina and team of authors

Reviewer 2 Report
The authors reported at least 194 beetle species from 21 families and branchiopod crustaceans and their distribution from the Quaternary deposits in the lower current of Suzun River. The topic is interesting and the manuscript is well written. Some small problems are as follows. I would recommend accept with minor revisions.
Line 51: The “MIS 3” should be changed for” the Marine Isotope State 3 (MIS 3)”
Line 59: The “the Marine Isotope State 3 (MIS 3)” should be changed for “MIS 3”.
Line 71: In “Materials and Methods” part, how did the authors identify the species? It should be stated more detailed.
Line 137-140: “The remaining beetle families (Silphidae, Scarabaeidae, Tenebrionidae, Dytiscidae, Helophoridae, Elateridae, Brentidae, Scolytidae, Hydrophilidae, Hydraenidae, Leiodidae, Byrrhidae, Meloidae, Heteroceridae, Malachiidae, and Cerambycidae) are represented only by singletons.”
It is difficult to identify a specific species from a single specimen, please verify.
Author Response
Dear Reviewer, We express our sincere gratitude to you for reviewing our work and making very valuable comments. Your comments have been taken into consideration.
Line 51: The “MIS 3” should be changed for” the Marine Isotope State 3 (MIS 3)” - It was done.
Line 59: The “the Marine Isotope State 3 (MIS 3)” should be changed for “MIS 3”. - It was done.
Line 71: In “Materials and Methods” part, how did the authors identify the species? It should be stated more detailed. - We made the additions to the chapter.
Line 137-140: “The remaining beetle families (Silphidae, Scarabaeidae, Tenebrionidae, Dytiscidae, Helophoridae, Elateridae, Brentidae, Scolytidae, Hydrophilidae, Hydraenidae, Leiodidae, Byrrhidae, Meloidae, Heteroceridae, Malachiidae, and Cerambycidae) are represented only by singletons.”
It is difficult to identify a specific species from a single specimen, please verify.
- Paleontologists often work with single specimens. Sometimes it is difficult, but the more significant factor is the safety of the material. An explanatory phrase has been added to the methods chapter.
Wit best wishes, Anna Gurina and team of authors
